# Midbrain ghrelin receptor signalling regulates binge drinking in a sex specific manner

Amy J. Pearl[1,7], Xavier J. Maddern[1,2,7], Paulo Pinares-Garcia[1], Lauren T. Ursich[1,2], Roberta G. Anversa[1,2], Arnav Shesham[1,3], Robyn M. Brown[1,4], Felicia M. Reed[3], William J. Giardino [5,6], Andrew J. Lawrence [1,2] & Leigh C. Walker[1,2] ✉

Risky drinking rates are rising, particularly in women, yet sex as a biological variable has only recently gained traction. The centrally projecting Edinger-Westphal (EWcp) nucleus has emerged as a key regulator of alcohol consumption. Here we found that EWcp[peptidergic] cells reduce binge drinking specifically in female mice. We show this effect is mediated by the ghrelin receptor (GHSR), with EWcp[peptidergic] inhibition blocking ghrelin-induced drinking and *Ghsr* knockdown in EWcp[peptidergic], but not EWcp[glutamatergic] or ventral tegmental area cells, reducing binge drinking in females, independent of circulating sex hormones. Female mice showed higher EWcp *Ghsr* expression, and EWcp[peptidergic] neurons were more sensitive to ghrelin. Moreover, intra-EWcp delivery of GHSR inverse agonist and antagonist reduced binge drinking, suggesting direct actions of ghrelin. These findings highlight the EWcp as a critical mediator of excessive alcohol consumption via GHSR in female mice, offering insights into the ghrelin system's role in alcohol consumption.

Whilst men have historically had higher rates of alcohol use, misuse and alcohol use disorder (AUD) compared to women, statistics from the US suggest these rates have converged significantly over recent decades, primarily driven by an increase in risky drinking and AUD rates in women[1–3]. The existence of sex differences in alcohol consumption have been widely detailed[4–6]. However, sex as a biological variable has only recently gained traction as a critical factor, with most preclinical research and drug development identifying and testing therapies primarily in male subjects[7]. Thus, prioritising research on the neural mechanisms contributing to AUD in females and understanding sex differences is critical for developing effective treatment strategies.

Binge drinking, defined as a pattern of alcohol consumption that raises blood alcohol levels to 0.08 g/dl (NIAAA) is an early step in the progression of AUD[8]. The putative circuitry mediating this form of excessive alcohol consumption includes the understudied Edinger-Westphal (EW)[9–12]; a structure dense in neuropeptide expression, including cocaine and amphetamine regulated transcript (CART), urocortin (Ucn1) and cholecystokinin (CCK) and sex steroid hormone receptors (estrogen ERα, ERβ and progesterone mPRs)[13,14]. The evolutionarily conserved EW is separated into two main divisions, the preganglionic (EWpg), a population of cholinergic neurons that project to the ciliary ganglion, controlling oculomotor coordination, and the centrally projecting (EWcp), which consists of densely clustered neurons that project through the CNS[13–16]. Recent advancements have shown two major, but discrete populations of neurons within the EWcp exist, and can be broadly defined as peptidergic (expressing CART, Ucn1, CCK) or glutamatergic (expressing vGlut2)[17,18]; and these neurons are critical for regulating energy homoeostasis[19] and anxiety responses[18] across sexes, and preparatory nesting in pregnant female mice[17]. Further, while studies have shown a critical role for this nucleus

[1]Florey Institute of Neuroscience and Mental Health, Parkville, VIC 3052, Australia. [2]Florey Department of Neuroscience and Mental Health, University of Melbourne, Melbourne, VIC 3052, Australia. [3]Biomedicine Discovery Institute and Department of Physiology, Monash University, Clayton, VIC, Australia. [4]Department of Biochemistry and Pharmacology, University of Melbourne, Melbourne, VIC 3052, Australia. [5]Dept. of Psychiatry and Behavioural Sciences, Stanford University School of Medicine, Stanford, CA 94305-5453, USA. [6]Wu Tsai Neurosciences Institute, Stanford University School of Medicine, Stanford, CA 94305-5453, USA. [7]These authors contributed equally: Amy J. Pearl, Xavier J. Maddern. ✉e-mail: leigh.walker@florey.edu.au

in alcohol consumption[9,10,13,20,21], this has not been well characterised across sexes. Furthermore, while a role for the glutamatergic EWcp in regulating alcohol consumption was recently identified[21], a unifying model for how the peptidergic EWcp cells regulate alcohol consumption remains incomplete.

Interestingly the EWcp expresses a number of receptors involved in feeding regulation, including dense expression of the ghrelin receptor (growth hormone secretagogue receptor, GHSR)[22,23]. Indeed, EWcp GHSR expression is striking, and second only to GHSR expression in the well-studied hypothalamic arcuate nucleus (Arc) in mice[22]. GHSR is activated by its cognate ligand, ghrelin, a 28-amino acid peptide secreted from the stomach, and can be inhibited by the endogenous competitive antagonist/inverse agonist liver-expressed antimicrobial peptide 2 (LEAP-2)[24–26]. While this system emerged as an important regulator of energy balance and body weight homoeostasis[27] the evidence of the relationship between ghrelin/GHSR and alcohol consumption/seeking is growing (see[28] for review). Preclinical and clinical studies have highlighted a bidirectional relationship between endogenous ghrelin levels, or exogenous administration, with alcohol consumption/craving[28–35], and GHSR antagonists and inverse agonists have shown efficacy in reducing alcohol consumption when administered both peripherally and centrally[32,36–39]. Further, recent clinical trials have shown safety, tolerability, and some efficacy of a GHSR inverse agonist, PF-5190457, to reduce craving in heavy drinking individuals[40,41].

The central mechanism(s) that underpin ghrelin/GHSRs actions in alcohol consumption and craving are not well understood. Preclinical studies point to a role for GHSR signalling in the laterodorsal tegmental area (LDTg) and ventral tegmental area (VTA)[32,36,42]. Interestingly, peripheral administration of the GHSR antagonist, JMV2959, reduces alcohol-induced activation of the EWcp, but not VTA or arcuate nucleus (ARC)[43], suggesting the EWcp may mediate some of the central actions of GHSRs ability to reduce alcohol consumption.

While most studies examining the ghrelin system have been conducted exclusively in male subjects, sex differences in the effects of ghrelin/GHSR signalling are emerging[44,45]. Several studies have reported higher circulating levels of ghrelin in females than in males[46–48], and female mice are more sensitive to ghrelin's effects on feeding[44]. Here we identified sex differences in EWcp CART-expressing cells driving binge-like alcohol consumption. We then tested the hypothesis that GHSR signalling in the EWcp was responsible for these sex specific actions. We found sex differences in *Ghsr* mRNA expression and response to ghrelin in the EWcp, and functionally determined that GHSR signalling on CART-expressing cells in the EWcp drives alcohol consumption through both ligand dependent and independent actions. Together, these data represent a mechanism by which ghrelin acts to regulate binge-like alcohol consumption in a sex specific manner.

## Results

### EW CART cells mediate binge drinking in a sex specific manner

Given the known role of the EWcp in alcohol consumption, and recent finding that chemogenetic activation of EWcp vGlut2 cells reduces alcohol consumption in male and female mice[21], we sought to determine the role of a distinct population of peptidergic EWcp cells in binge drinking. To explore EWcp activation by alcohol, mice were perfused directly after 2 h binge session, or time/age/sex matched naïve controls (Fig. 1A). Female mice showed greater alcohol intake than males ($p = 0.0217$, Fig. 1B). Brains were processed for Fos and CART immunoreactivity to identify neural activation in the EWcp after alcohol. Alcohol increased the % of CART+Fos-positive cells in the EWcp of female ($p < 0.001$) and male ($p = 0.0152$) mice. Further, female alcohol binge mice showed greater % of CART+Fos positive cells than male mice ($p = 0.0132$, Fig. 1C, D). A positive correlation between % CART+Fos cells and alcohol intake (g/kg) was also observed

($R^2 = 0.3877$, $p = 0.0132$, Fig. 1E). A significant increase was observed in % Fos/CART-positive cells in females (naïve vs. alcohol, $p = 0.0187$), but not males (naïve vs. alcohol $p = 0.4490$), nor between sexes after alcohol ($p = 0.4232$, Fig. 1F). Further, no differences were noted in % of Fos+CART-negative cells in female (naïve vs. alcohol, $p = 0.2222$) or male (naïve vs. alcohol, $p = 0.2215$) mice, or between sex differences after alcohol (female vs. male $p = 0.6353$; Fig. 1G).

To explore a functional role of these peptidergic neurons, we used a chemogenetic approach in iCART-Cre mice (Fig. 1H–J). No differences were observed in weight (female $p = 0.8869$, male $p = 0.8393$; Fig. S1B, C) or training alcohol consumption (female $p = 0.6999$, male $p = 0.9184$; Fig. S3D, E) between control (mCherry) and mice with hM4Di (inhibitory) DREADD injected in the EWcp. Female mice with mCherry virus showed no difference between CNO and saline administration in either cumulative (main effect CNO, $p = 0.1891$, Fig. 1K) or total alcohol consumption ($p = 0.3727$; Fig. 1L), suggesting no off-target effects of CNO. However, female mice expressing hM4Di DREADDs showed a specific reduction in alcohol consumption following CNO administration in cumulative (main effect CNO, $p = 0.0020$; Fig. 1M) and total alcohol intake ($p = 0.0097$, Fig. 1N). By contrast, male mice with either mCherry or hM4Di showed no difference in binge alcohol consumption during test sessions between saline and CNO treatment in both cumulative intake (mCherry, $p = 0.4723$; hM4Di, $p = 0.9517$) or total consumption (mCherry, $p = 0.8496$; hM4Di, $p = 0.7385$; Fig. 1O–R). When comparing Δ intake between males and females, we observed a specific reduction in alcohol consumption across time in female CNO treated mice compared to males (main effect of sex, $p = 0.0133$; Fig. S3A, B), highlighting the sex specific effects of EW$^{CART}$ chemogenetic inhibition.

During the binge drinking test, female hM4Di mice treated with CNO showed a specific reduction in calories consumed from alcohol ($p = 0.0002$; Fig. S1H), but not food ($p = 0.8462$; Fig. S1I), nor were effects seen in male mice for alcohol ($p = 0.8063$, Fig. S1J) or food ($p = 0.8295$; Fig. S1K). Other behaviours were also assessed to determine the specificity of EWcp peptidergic cell inhibition on binge drinking. No difference in water or food intake were observed in female mice (water $p = 0.2892$; food, $p = 0.8194$; S1L-N), or male mice (water, $p = 0.0865$; food $p = 0.6008$; Fig. S1O, P), or saccharin preference in either sex (female $p = 0.7528$, male $p = 0.3317$; Fig S1Q–S). Further, no difference in anxiety-like behaviours were observed in either sex during the light-dark box procedure ($p = 0.2241$, male $p = 0.9181$; Fig. S2A–G), no locomotor deficits were observed (female, $p = 0.8082$, male, $p = 0.9290$; Fig. S2H, I), nor any differences in body temperature (female, $p = 0.7971$; male, $p = 0.2455$; Fig. S2J, K). Together, these data indicate that EWcp peptidergic neurons specifically reduce alcohol consumption in female mice, without altering other behaviours.

### EW GHSR signalling mediates binge drinking in a sex specific manner

The EW expresses various receptors for feeding-related peptides, including the ghrelin receptor (GHSR), which has been linked to alcohol use[28]. However, the role of these receptors within the EWcp remains unexplored. To assess whether ghrelin signalling in the EWcp regulates alcohol consumption we next trained mice expressing hM4Di DREADD receptors on EW$^{CART}$ cells on a "drinking in the light" procedure, which gave limited access to alcohol for 2 h, 3 times weekly within the light phase when ghrelin levels are low (Fig. 2A–C). Ghrelin administration increased alcohol consumption in female (cumulative intake, saline/saline vs. saline/ghrelin $p = 0.0447$; trend in total intake, saline/saline vs. saline/ghrelin $p = 0.0592$, Fig. 2D, E), but not male mice (cumulative intake, main effect treatment $p = 0.9124$, total intake, $p = 0.9614$; Fig. 2F, G). In female mice, EW$^{CART}$ DREADD inhibition prevented this escalation in alcohol consumption compared to saline + ghrelin treated hM4Di DREADD mice (cumulative intake, saline/ghrelin

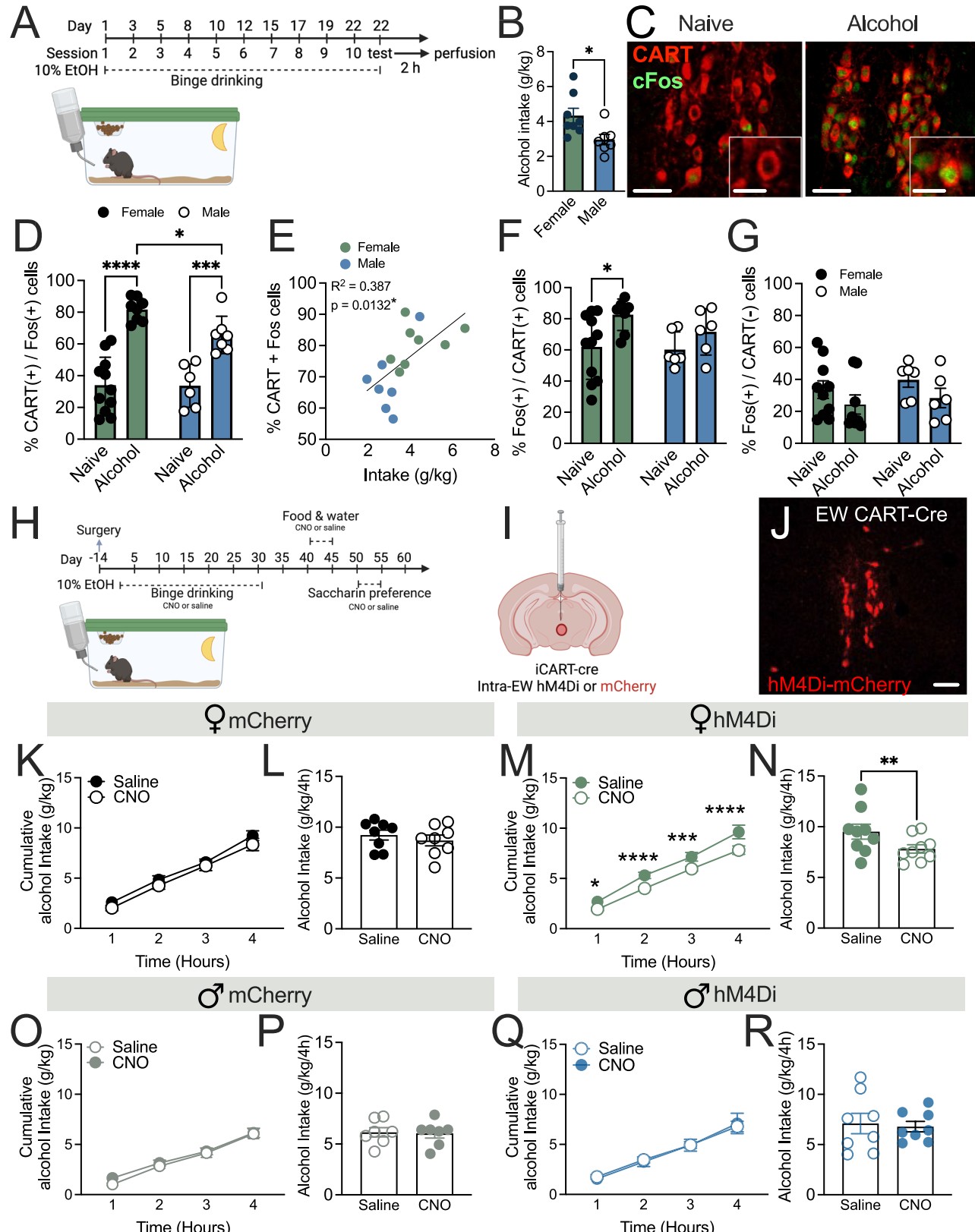

vs. CNO/ghrelin $p = 0.0209$, saline/saline vs. CNO/ghrelin $p > 0.9999$; total intake, saline/ghrelin vs. CNO/ghrelin $p = 0.0404$, saline/saline vs. CNO/ghrelin $p > 0.9999$; Fig. 2D, E). When comparing Δ intake between males and females we observed a specific reduction in ghrelin-induced alcohol consumption in female CNO treated mice compared to males (trend towards main effect of sex, $p = 0.0790$; Fig. S3C, D), highlighting

the sex specific effects of $EW^{CART}$ chemogenetic inhibition on ghrelin induced alcohol drinking.

To specifically probe the direct effect of $EW^{GHSR}$ in binge drinking, we disrupted *Ghsr* gene expression in adult mouse brain using an shRNA knockdown approach (Fig. 2H, I). RT-qPCR validation showed a specific knockdown of *Ghsr* in the EW ($p < 0.0001$), but not the

**Fig. 1 | Edinger Wesphal (EW)CART cells mediate alcohol binge drinking in female, but not male mice. A** Schematic of experimental timeline. **B** Female mice consumed more alcohol than males ($p = 0.0217$; n's=8 F, 7 M). **C** Representative image of CART+Fos labelling in the EWcp. **D** Female and male mice showed increased % CART+Fos expression in the EWcp after alcohol consumption compared to naïve controls (female $p < 0.001$, male $p = 0.0152$), with female alcohol mice showing greater %CART+Fos than male counterparts ($p = 0.0132$; n's=11 F naïve, 8 F alcohol, 6 M naïve, 7 M alcohol). **E** Alcohol consumption showed a positive correlation with %CART+Fos expression in the EWcp ($R^2 = 0.3877$, $p = 0.0132$; $n = 15$ [8 F, 7 M pooled]). **F** Female, but not male, mice showed increased %Fos+CART-positive expression in the EWcp (naïve vs. alcohol female $p = 0.0187$ [$n = 11$ naïve, 8 alcohol]; male $p = 0.4232$ [$n = 6$ naïve, 6 alcohol]), but neither sex showed differences in (**G**) % Fos+CART-negative (female $p = 0.2222$; male $p = 0.2215$) nor differences between sexes in alcohol groups ($p = 0.6353$; n's=10 F naïve, 8 F alcohol, 6 M naïve, 6 M alcohol). **H** Schematic of experimental timeline for chemogenetic targeting, **I** viral approach and **J** representative image of viral transduction of EWCART cells. Female mice injected with control virus showed no changes in alcohol consumption

following CNO administration in **K** cumulative intake ($p = 0.1891$; $n = 8$) or **L** total alcohol consumption ($p = 0.3727$; $n = 8$). However, female mice injected with hM4Di virus showed a reduction in alcohol consumption following CNO administration in **M** cumulative intake across time ($p = 0.0020$; $n = 9$) and **N** total alcohol consumption ($p = 0.0097$; $n = 9$). Male mice that were injected with control virus showed no changes in alcohol consumption following CNO administration in **O** cumulative intake ($p = 0.4723$; $n = 7$) or **P** total alcohol consumption ($p = 0.8496$; $n = 7$). Male mice injected with hM4Di virus showed no change in **Q** cumulative ($p = 0.9517$; $n = 8$) or **R** total alcohol intake ($p = 0.7385$; $n = 8$). Data expressed as mean ± SEM. Panel **B, L, N, P, R** analysed by two-sided unpaired $t$-test, Panel **D, F, G, K, M, O, Q** analysed by two-way ANOVA with Bonferroni *post-hoc*, Panel **E** analysed by linear regression. *$p < 0.05$, **$p < 0.01$, ***$p < 0.001$, ****$p < 0.0001$. Scale bar= 100 μm (C overview), 40 μm (C overlay), 200 μm **J**. Full statistics in Table S3 and source data are provided as a Source Data File. Created in BioRender. Walker, L. (2025) https://BioRender.com/u78n049 & https://BioRender.com/l95j352 [Agreement #AI27UEQ6WA & UC27UEQHWA].

neighbouring VTA ($p = 0.4256$), in male and female mice (Fig. 2J). Female mice injected with Ghsr-ShRNA during training exhibited decreased alcohol consumption compared to mice treated with scram (Sc)-ShRNA, which showed no change (session x treatment interaction, $p = 0.0170$, Fig. 2K). This reduction persisted during the test when compared to Sc-ShRNA treated mice (cumulative, main effect of treatment, $p = 0.0414$; total intake $p = 0.0281$, Fig. 2L, M) and EW *Ghsr* expression positively correlated with binge drinking behaviour ($R^2 = 0.4332$, $p = 0.0142$, Fig. 2N). No effect of Ghsr-ShRNA was observed in male mice (cumulative intake main effect of treatment, $p = 0.6473$, total intake, $p = 0.8315$; Fig. 2O–Q), nor did EW *Ghsr* expression correlate with alcohol consumption in males ($R^2 = 0.0036$, $p = 0.9532$; Fig. 2R). When comparing Δ intake between males and females, we observed a specific reduction in alcohol consumption in female shRNA treated mice compared to males (main effect of sex, $p = 0.0093$; Fig. S3E, F), highlighting the sex specific effects of EW *Ghsr* knockdown in binge drinking.

Further, given the expression of GHSR in the neighbouring VTA, and its known actions in modulating the mesocorticolimbic dopamine system[42,49], we also assessed the impact of *Ghsr* knockdown within the VTA of both male and female mice. RNAscope showed *Ghsr* mRNA is expressed on both ~67% of DAT+ and 33% DAT- cells (Fig. S4A, B). C57BL6J mice injected with Ghsr-ShRNA showed no difference in alcohol consumption during training (Δ intake from pre-surgery, females $p = 0.6947$; Fig. S4E; males $p = 0.1548$; Fig. S4G) or cumulative intake during test (females: $p = 0.9765$; Fig S4F; males: $p = 0.0993$; Fig. S4H). We further characterised whether VTA dopamine cells may be more specifically involved by injecting a Cre-dependent Ghsr-ShRNA into the VTA of DAT-Cre mice (Fig. S4I, J). DAT-Cre mice injected with Ghsr-ShRNA showed no difference in alcohol consumption during training (Δ intake from pre-surgery; female, $p = 0.9697$; Fig. S4K; male, $p = 0.6735$; Fig. S4G) or cumulative intake during test (female, $p = 0.9765$; Fig. S4L; male $p = 0.7552$; Fig. S4H), suggesting interruption of the ghrelin receptor in the VTA does not alter alcohol consumption in either sex. Further, when comparing Δ intake between males and females we observed no sex differences in alcohol consumption in either Ghsr-ShRNA in the VTA of C57BL6J mice (main effect of sex, $p = 0.7139$; Fig. S3G, H), or DAT-Cre mice (main effect of sex, $p = 0.8449$; Fig. S3I, J).

### Ghsr knockdown in the EW is not altered by circulating sex hormones

To determine if circulating hormones mediate the effect of *Ghsr* knockdown observed in female mice, we next conducted ovariectomy (OVX) or SHAM surgeries in female mice before disrupting *Ghsr* gene expression (Fig. 3A, B). RT-qPCR validation showed specific knockdown of *Ghsr* in the EW in female SHAM ($p = 0.0040$) and OVX

($p = 0.0075$) mice (Fig. 3C). *Ghsr*-ShRNA knockdown reduced alcohol consumption during post-surgery training (Δ alcohol consumption) independent of OVX (SHAM, S0 vs. S1 $p = 0.003$, S0 vs. S2 $p = 0.0302$, S0 vs. S3 $p = 0.0066$, S0 vs. S4 $p = 0.0057$, S0 vs. S5 $p = 0.3250$, S0 vs. S6 $p = 0.0339$; OVX, S0 vs S1 $p = 0.0005$, S0 vs. S2 $p = 0.0050$, S0 vs. S3 $p = 0.0017$, S0 vs. S4 $p = 0.0008$, S0 vs. S5 $p = 0.0018$, S0 vs. S6 $p = 0.0009$; Fig. 3D), as no difference was observed between SHAM and OVX mice (main effect treatment $p = 0.3802$, Fig. 3D), nor was any effect observed during test (cumulative intake $p = 0.9530$; total intake, $p = 0.5458$; Fig. 3E, F). Uterine weight was significantly reduced in OVX mice ($p = 0.0012$, Fig. 3G), demonstrating the OVX surgery was effective.

### Sex differences in EW Ghsr expression and response to ghrelin

Recent studies have identified two distinct populations of EWcp cells, with distinct neurochemistry (peptidergic vs. glutamatergic)[17,18]. While dense expression of GHSR has been reported within the EW[22,23], its distribution across newly defined populations has not been well characterised. Therefore, we conducted RNAscope, RT-qPCR and electrophysiology to explore *Ghsr* expression and function within the EWcp across sexes. Our data show dense expression of *Ghsr* in 99% of peptidergic (*Cartpt* + ), and lesser expression in ~40% of glutamatergic (*Slc17a6* + ) cells (Fig. 4A, C). Quantification of RNAscope showed a trend towards increased number of *Ghsr* ($p = 0.0922$), but not *Cartpt* ($p = 0.4598$), or *Slc17a6* (*vGlut2*; $p = 0.2668$) expressing cells in female mice, compared to male counterparts (Fig. 4D). In male mice, 42% of *Ghsr*+ cells were identified as peptidergic, with 49% identified as glutamatergic, and the remaining 9% exhibiting neither phenotype. In female mice, 34% of *Ghsr*-positive cells exhibited a peptidergic phenotype, while 61% were identified as glutamatergic, and *Ghsr* expression observed in isolation on 5% of cells (Fig. 4E). In line with this, RT-qPCR analysis showed a trend towards greater *Ghsr* expression in female mice ($p = 0.0525$, Fig. 4G). Further analysis also showed greater expression of the estrogen receptor 1 mRNA ($p = 0.0039$, ERα, *Esr1*, Fig. 4H), and estrogen receptor 2 mRNA ($p = 0.0274$, ERβ, *Esr2*, Fig. 4I), but not membrane bound progesterone receptors (mPRγ, *Parq5*, $p = 0.2273$; mPRβ, *Parq8*, $p = 0.8947$; Fig. 4J, K) in female mice compared to male counterparts. Whole cell slice recordings revealed no differences in basal electrophysiological properties between sexes (Fig. S5). Further, bath application of ghrelin increased firing rate of EWcp peptidergic cells in both female and male mice (female, $p < 0.0001$, male, $p = 0.0005$; Fig. 4M). However, EWcp peptidergic cells in female mice were more sensitive to ghrelin, showing a higher firing rate in response to ghrelin compared to male counterparts ($p = 0.0438$, Fig. 4N). A significant decrease in spike amplitude was also observed following bath application of ghrelin in male and female mice (female, $p < 0.001$, male, $p < 0.001$; Fig. 4O); however, there was no

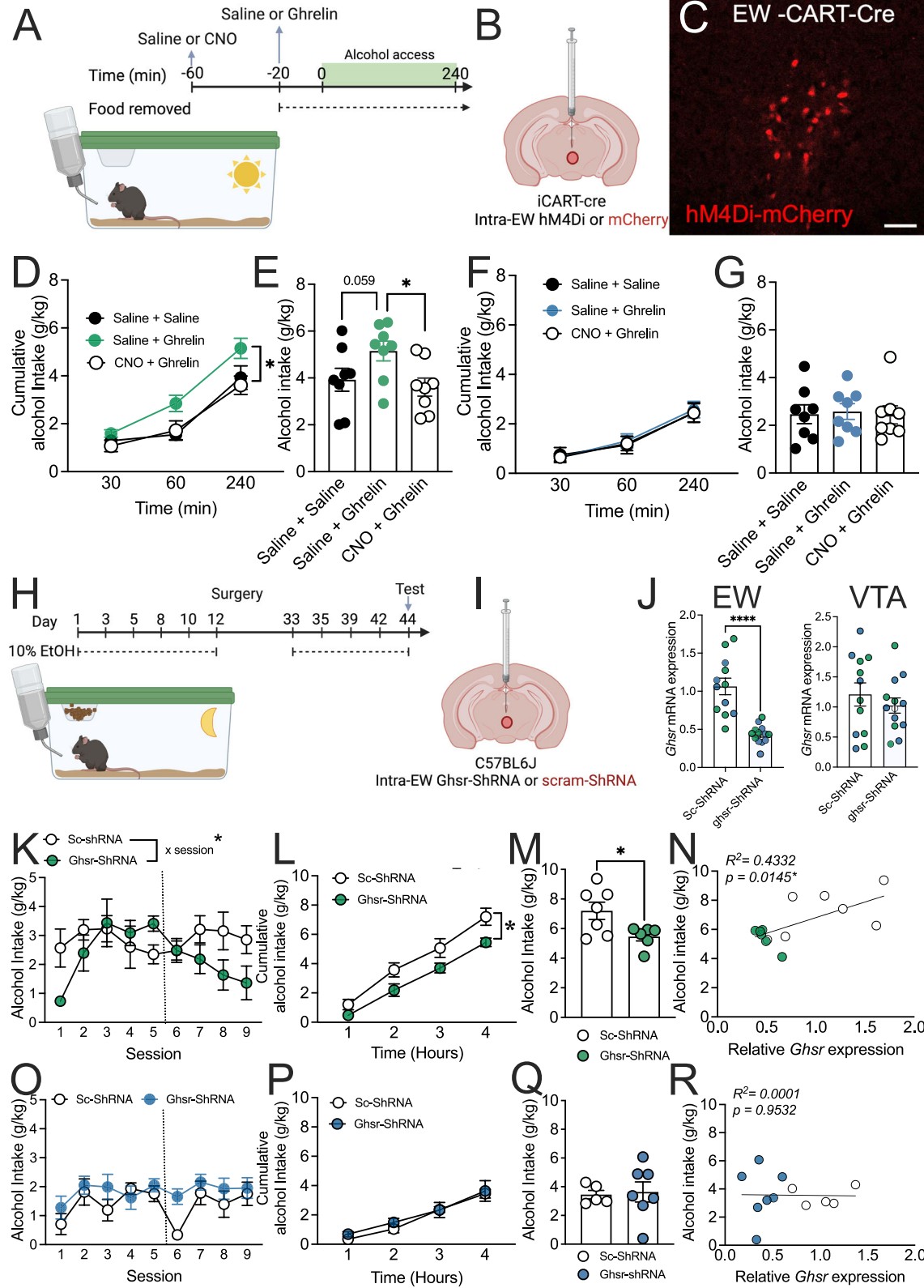

difference between sexes ($p = 0.3581$; Fig. 4P). Bath ghrelin application did not alter Δ RMP ($p = 0.9175$; Fig. S6A), Δ Rm ($p = 0.1305$; Fig. S6B), Δ Tau ($p = 0.2118$; Fig. S6C), or Δ Cm ($p = 0.1576$; Fig. S6D), but an increased Δ sag current was observed in female mice ($p = 0.0437$, Fig. S6E), with a trend towards increased Δ sag ratio also observed in females ($p = 0.0658$; Fig. S6F). Representative traces are shown in Fig. 4Q, R for males and females respectively.

## Cell-type specific role of GHSR in regulating alcohol consumption in female mice

After confirming *Ghsr* is expressed across EW[vGlut2] and EW[CART] cells, we next assessed whether *Ghsr* expressed specifically on EW[CART] or EW[vGlut2] cells functionally drives alcohol consumption in female mice. To specifically target and disrupt *Ghsr* gene expression in EW subpopulations of the adult mouse brain we used a Cre dependent

**Fig. 2 | EWcp ghrelin/GHSR signalling regulates alcohol consumption in female, but not male mice. A** Experimental timeline, **B** viral approach and **C** representative image of viral transduction. Ghrelin administration increased **D** cumulative ($p = 0.0447$; $n = 8$) and **E** trend towards increased total alcohol consumption in female mice ($p = 0.0592$), which was prevented by CNO pre-treatment (cumulative intake, saline/ghrelin vs. CNO/ghrelin $p = 0.0209$, saline/saline vs. CNO/ghrelin $p > 0.9999$; total intake, saline/ghrelin vs. CNO/ghrelin $p = 0.0404$, saline/saline vs. CNO/ghrelin $p > 0.9999$; $n = 8$). Ghrelin administration did not alter **F** cumulative ($p = 0.9124$; $n = 8$) or **G** total alcohol consumption in males ($p = 0.9614$; $n = 8$). **H** Experimental timeline, **I** and viral approach. **J** qPCR showed a specific knockdown in *Ghsr* mRNA in the EW ($p < 0.0001$), but not VTA ($p = 0.4256$; n's=11 Sc-ShRNA [7 F, 5 M pooled], 13 *Ghsr*-ShRNA [6 F, 7 M pooled]; males=blue, females=green). **K** Training pre- and post-ShRNA showed a reduction in alcohol consumption in female *Ghsr*-ShRNA mice (session x treatment interaction $p = 0.0170$; n's=7 Sc-ShRNA, 6 *Ghsr*-ShRNA). A reduction in alcohol consumption was observed in female *Ghsr*-ShRNA mice during a 4-h test in **L** cumulative ($p = 0.0414$; n's=7 Sc-ShRNA, 6 *Ghsr*-ShRNA) and **M** total alcohol intake ($p = 0.0281$; n's=7 Sc-ShRNA, 6 *Ghsr*-ShRNA). **N** Alcohol consumption positively correlated with *Ghsr* expression in

female mice (open circles=*Ghsr*-ShRNA, closed circles=Sc-ShRNA; $R^2 = 0.4332$, $p = 0.0142$; $n = 13$ [7 Sc-ShRNA, 6 *Ghsr*-ShRNA pooled]). **O** Training pre- and post-shRNA showed no difference in alcohol consumption in male mice (treatment x time $p = 0.4380$; n's=5 Sc-ShRNA, 7 *Ghsr*-ShRNA). No difference in alcohol consumption was also observed in male *Ghsr*-ShRNA mice during 4-h test in **P** cumulative ($p = 0.6473$; n's=7 Sc-ShRNA, 6 *Ghsr*-ShRNA) and **Q** total alcohol intake ($p = 0.8315$; n's=7 Sc-ShRNA, 6 *Ghsr*-ShRNA). **R** Alcohol consumption did not correlate with *Ghsr* expression in males (open circles=Ghsr-ShRNA, closed circles=Sc-ShRNA; $R^2 = 0.0036$, $p = 0.9532$; $n = 13$ [7 Sc-ShRNA, 6 *Ghsr*-ShRNA pooled]). Data expressed as mean ± SEM. Panel **D, F, K, L, O, P** analysed by RM two-way ANOVA, panel **E, G** analysed by RM one-way ANOVA, Bonferroni *post-hoc* used when significant main effects observed for ANOVAs, Panel **J, M, Q** analysed by two-tailed unpaired *t*-test, Panel **N, R** analysed by linear regression. *$p < 0.05$, **$p < 0.01$, ***$p < 0.001$, ****$p < 0.0001$. Scale bar=200 μm. Full statistics in Table S3 and source data are provided as a Source Data File. Created in BioRender. Walker, L. (2025) https://BioRender.com/x51b823 & https://BioRender.com/c61p137 [Agreement #AI27UEQ6WA & UC27UEQHWA].

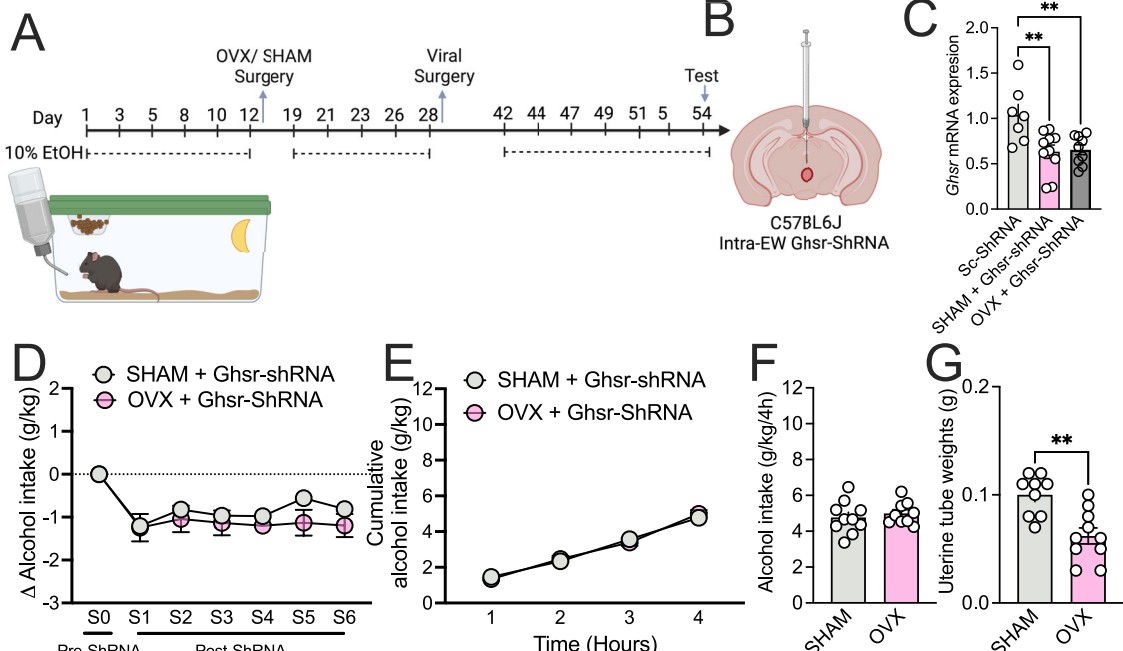

**Fig. 3 | Circulating sex hormones does not impact the effect of *Ghsr* knockdown in female mice. A** Schematic of experimental timeline, **B** details of viral approach and **C** qPCR validation of *Ghsr* mRNA expression in the EW following ShRNA knockdown in SHAM ($p = 0.0040$) and OVX ($p = 0.0075$) compared to Sc-ShRNA controls (n's=7 Sc-ShRNA, 10 SHAM+*Ghsr*-ShRNA, 10 OVX+*Ghsr*-ShRNA). **D** Training pre- and post-ShRNA knockdown showed a reduction in alcohol consumption in female mice with SHAM or OVX and ShRNA knockdown compared to their baseline (S0) (SHAM, S0s vs. S1 $p = 0.003$, S0 vs. S2 $p = 0.0302$, S0 vs. S3 $p = 0.0066$, S0 vs. S4 $p = 0.0057$, S0 vs. S5 $p = 0.3250$, S0 vs. S6 $p = 0.0339$; OVX, S0 vs S1 $p = 0.0005$, S0 vs. S2 $p = 0.0050$, S0 vs. S3 $p = 0.0017$, S0 vs. S4 $p = 0.0008$, S0 vs. S5 $p = 0.0018$, S0 vs. S6 $p = 0.0009$), but no difference between OVX and SHAM groups ($p = 0.3802$) (n's=10 SHAM + *Ghsr*-ShRNA, 10 OVX+*Ghsr*-ShRNA). No difference in alcohol consumption was also observed in SHAM or OVX *Ghsr*-ShRNA treated mice during an

extended 4-h test in both **E** cumulative ($p = 0.9530$; n's=10 SHAM + *Ghsr*-ShRNA, 10 OVX + *Ghsr*-ShRNA) or **F** total alcohol intake ($p = 0.5458$; n's=10 SHAM+*Ghsr*-ShRNA, 10 OVX+*Ghsr*-ShRNA). **G** A reduction in uterine weights were observed in OVX mice compared to SHAM ($p = 0.0012$; n's=10 SHAM+*Ghsr*-ShRNA, 10 OVX +*Ghsr*-ShRNA). Data expressed as mean ± SEM. Panel **C** analysed by one-way ANOVA, Panel **D, E** analysed by two-way ANOVA. Bonferroni *post-hoc* used when significant main effects observed for ANOVAs, Panel **F, G** analysed by two-tailed unpaired student *t*-test. **$p < 0.01$; $^\S p < 0.05$; $^{\S\S}p < 0.01$, $^{\S\S\S}p < 0.001$ compared with S0 in SHAM *Ghsr*-ShRNA mice; $^\#p < 0.05$, $^{\#\#}p < 0.01$, $^{\#\#\#}p < 0.001$ compared with S0 in OVX *Ghsr*-ShRNA mice. Full statistics in Table S3 and source data are provided as a Source Data File. Created in BioRender. Walker, L. (2025) https://BioRender.com/l24u000 [Agreement #ST27UEQPB7].

ShRNA approach in iCART-Cre (Fig. 5A–C) and vGlut2-Cre (Fig. 5G–I) mice. In iCART-Cre mice, *Ghsr*-ShRNA knockdown reduced alcohol consumption during post-surgery training (Δ alcohol consumption), compared to Sc-ShRNA treated mice (main effect of treatment, $p = 0.0082$, but no significant *post hoc* differences; Fig. 5D). This reduction persisted during the test for both cumulative intake $p = 0.0044$, 1 h $p = 0.2717$, 2 h $p = 0.0812$, 3 h $p = 0.0247$, 4 h ($p = 0.0063$) and total intake ($p = 0.0015$; Fig. 5E, F). No effect of

*Ghsr*-ShRNA was observed in vGlut2-Cre mice compared to scram-ShRNA controls during post-surgery training (main effect of treatment $p > 0.9588$; Fig. 5J), or test (cumulative intake $p = 0.3747$; total intake p = 0.2293; Fig. 5K, L). We also assessed selective *Ghsr* knockdown on sucrose consumption. *Ghsr*-ShRNA knockdown on EW$^{CART}$ cells did not alter sucrose consumption compared to Sc-ShRNA treated mice (cumulative intake $p = 0.8302$; total intake $p = 0.7767$; Fig. S7).

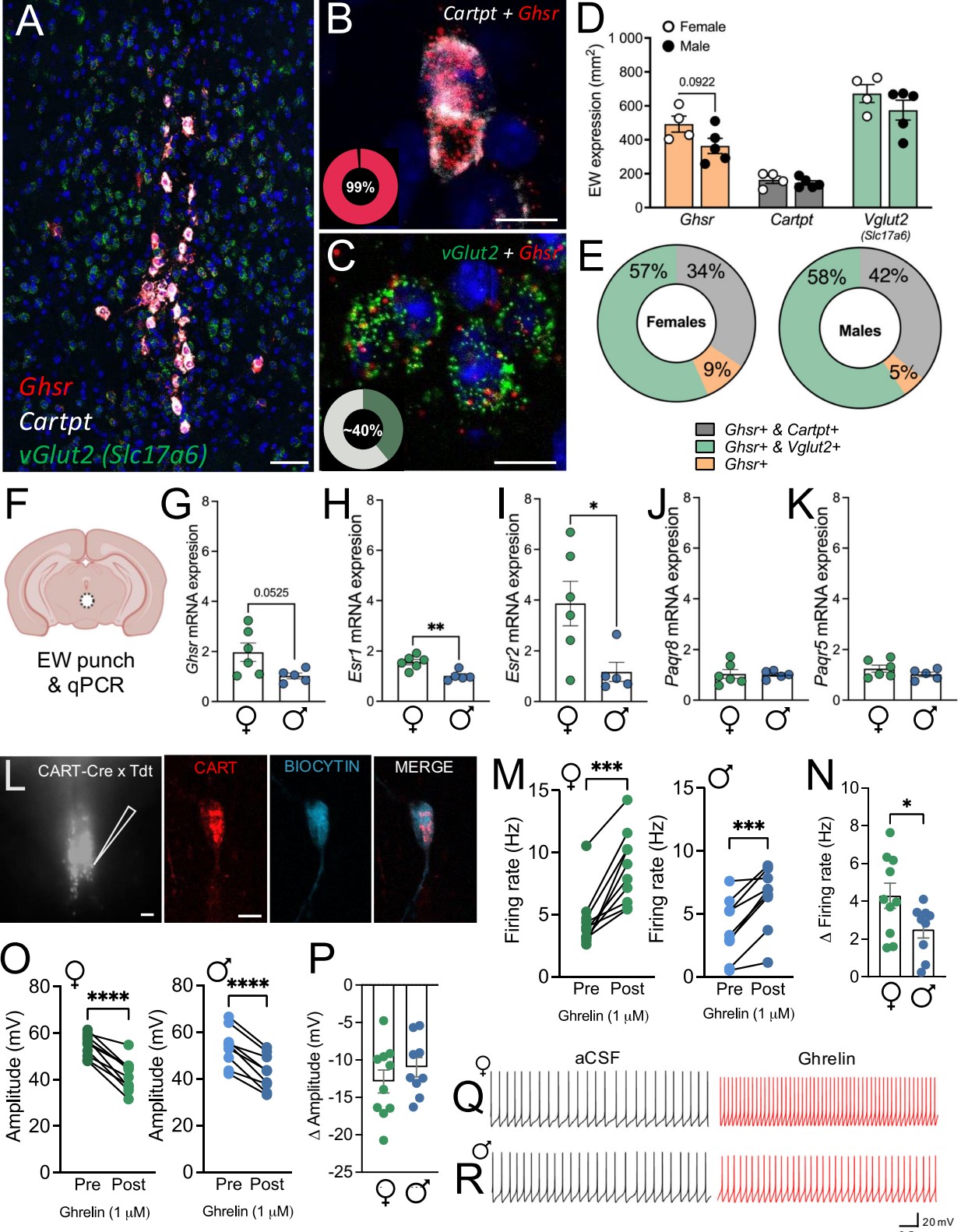

## GHSR1a constitutive and ligand dependent activity drive alcohol consumption in female mice

The *Ghsr* gene encodes two transcripts for GHSR1a and GHSR1b; GHSR1a is a 7-transmembrane GPCR and the cognate receptor for the peptide hormone ghrelin[50], whereas the truncated GHSR1b (lacking transmembrane domain 6 and 7) is not bound by ghrelin and is thought to instead regulate trafficking and signalling of GHSR1a within

the cell[51]. Interestingly, GHSR1a exhibits unusually high constitutive activity, with ~50% of its maximal capacity observed in the absence of ghrelin[52]. Therefore, we next sought to determine whether the actions of EWcp *Ghsr* knockdown were driven by surface-expressed GHSR1a, and if so whether these actions were via ligand dependent or independent mechanisms. Female mice were microinjected with either vehicle, JMV2959 (GHSR1a antagonist) or LEAP2 (GHSR1a inverse

**Fig. 4 | Sex differences in EWcp expression and activity. A** Overview of EWcp *Ghsr, Cartpt* and *vGlut2 (Slc17a6)* expression. **B** Representative microphotograph of *Ghsr* and *Cartpt* in the EWcp showed 99% of *Cartpt*-positive cells express *Ghsr* (n = 9 [4 F, 5 M pooled]). **C** Representative microphotograph of EWcp *Ghsr* and *vGlut2* expression showed 40% of *vGlut2*-positive cells express *Ghsr* (n = 9 [4 F, 5 M pooled]). **D** Cell density of *Ghsr, Cartpt* and *vGlut2* in male and female mice (*Ghsr*, $p = 0.0922$, *Cartpt*, $p = 0.4598$, *vGlut2*, $p = 0.2668$; n's=4 F, 5 M). **E** Proportions of *Ghsr*-positive cells that co-express *Cartpt, vGlut2* or neither, in male and female mice (n's=4 F, 5 M). **F** EW punch location. qPCR analysis showed **G** a trend towards increased *Ghsr* mRNA ($p = 0.0525$; n's=6 F, 5 M), increased estrogen receptor mRNA **H** *Esr1* (ERα, $p = 0.0039$; n's=6 F, 5 M) and **I** *Esr2* (ERβ, $p = 0.0274$; n's=6 F, 5 M), but no difference in progesterone receptor mRNA **J** *Parq5* (mPRγ, $p = 0.2273$; n's=6 F, 5 M), or **K** *Parq8* (mPRβ, $p = 0.8947$; n's=6 F, 5 M) in female compared to male mice. **L** Representative micrograph of EW tdTomato expression and confirmation of biocytin filled CART cells. **M** Firing frequency of EWcp$^{CART}$ neurons increased after application of 1 μM ghrelin in female (left, $p < 0.0001$; n = 10 cells/7 mice) and male (right, $p = 0.0005$ n = 9 cells/7 mice) mice. **N** Delta (Δ) firing rate (change in pre- vs. post-ghrelin administration) in male and female mice showed an increase in firing frequency in females in response to ghrelin compared to males ($p = 0.0438$; n's=10 cells/7 females, 9 cells/7 males). **O** Amplitude of EWcp$^{CART}$ neurons decreased after application of 1 μM ghrelin in female (left, $p < 0.001$, n = 10 cells/7 mice) and male (right, $p < 0.001$, n = 9 cells/7 mice) mice. **P** Δ amplitude showed no significant difference between sexes ($p = 0.3581$; n's=10 cells/7 females, 9 cells/7 males). Representative traces of EWcp$^{CART}$ cells following ghrelin application in **Q** female and **R** male mice. Data expressed as mean ± SEM. Panel **G–K, N, P** analysed by two-tailed unpaired students t-test, Panel **M, O** by two-tailed paired students t-test. *$p < 0.05$, **$p < 0.01$, ***$p < 0.001$, ****$p < 0.0001$. Scale bar=200μm **A, L**, 50μm **B, C, L**. Full statistics in Table S3 and source data are provided as a Source Data File. Created in BioRender. Walker, L. (2025) https://BioRender.com/o12v150 [Agreement #JQ27UESYS3].

agonist) directly within the EW, or adjacent (anatomical control) prior to the binge session (Fig. 6A–C). During training, mice with a cannula placed within the EW, or adjacent, showed similar consumption (main effect of treatment $p = 0.6379$; Fig. 6D, E) highlighting that cannula implantation per se did not alter this behaviour. Administration of JMV2959 or LEAP2 directly within the EW significantly reduced alcohol consumption (main effect of treatment, $p = 0.0058$; Fig. 6F). LEAP2 showed significant reduction across all time points (1 h $p = 0.0049$, 2 h $p = 0.0279$, 3 h $p = 0.0112$, 4 h $p = 0.0103$; Fig. 6F), while a significant effect of JMV2959 was noted only after the full 4 h session (1 h $p = 0.1430$, 2 h $p = 0.2508$, 3 h $p = 0.0632$, 4 h $p = 0.0265$, Fig. 6F). Total alcohol intake was also decreased after LEAP2 ($p = 0.0137$) and JMV2959 ($p = 0.0346$; Fig. 6G). This was specific to drug administration within the EW, with no effect observed when JMV2959 or LEAP2 were administered adjacent to the EW for either cumulative intake (main effect of treatment $p = 0.3797$; Fig. 6H) or total alcohol intake ($p = 0.5029$; Fig. 6I). Cannula placements are shown in Fig. 6J.

## Discussion

The EWcp is a critical node driving alcohol consumption[9,12,20], however the contributions of specific cell populations and mechanisms underpinning this behaviour until now have not been well defined. Our data identify a mechanism whereby ghrelin/GHSR1a signalling mediates binge drinking through the peptidergic cells in the EWcp, in a sex specific manner.

EWcp peptidergic cells are strongly activated by alcohol[9,12,13], in line with this we showed activation of EWcp peptidergic cells in response to binge drinking, and that selective chemogenetic inhibition of this population of cells reduces alcohol consumption. Interestingly, while alcohol activates the EWcp, specific comparisons of activation between sexes had not been assessed previously. However, the EWcp has recently been implicated as a connector hub linking regions activated by excessive alcohol consumption in female, but not male mice[53]. In line with this, we show while alcohol activates peptidergic EW cells in both sexes, a greater activation is observed in females, and this correlates to alcohol consumption. Further, when inhibiting the EWcp peptidergic cells, we observed a specific reduction of alcohol consumption in female, but not male mice. Recent studies have highlighted two prominent intermingled but separate populations of cells in the EWcp that either synthesise a number of neuropeptides (peptidergic – expressing CART, CCK, Ucn1), or are glutamatergic (expressing vGlut2)[17,18]. Previous studies have shown *Ucn1* gene knockdown in the EWcp reduces alcohol consumption in extended access procedures, but this effect did not differ between sexes[10]. Chemogenetic activation of neighbouring vGlut2 cells reduced alcohol consumption in male mice[21], however female mice were not examined. Our data extend these findings to report that chemogenetic inhibition

of peptidergic cells reduces alcohol consumption, however, our findings suggest this is specific to female mice, highlighting both behavioural distinction between these populations and potential sex differences in the role of EWcp peptidergic cells in driving alcohol consumption.

The EWcp expresses a number of sex steroid hormone receptors including those for estrogen (ERα & ERβ; Derks et al., 2007) and progesterone (mPRγ & mPRβ[17];) which may account for some sex dependent actions driven via the EWcp. Indeed, progesterone reduced firing frequency of EWcp peptidergic cells and ablation of EWcp peptidergic neurons reduced the responsiveness of female mice to progesterone-induced nesting behaviours[17]. We did not observe differences in progesterone receptor mRNA (mPRγ & mPRβ) in the EWcp between sexes in virgin mice, although significant differences in estrogen receptor mRNA expression (ERα & ERβ) were observed. Sex and developmental differences in ER binding have been reported within the hypothalamus, bed nucleus of the stria terminalis (BNST) and ARC of rodents[54]. Further, sex differences in ERα expression have been reported within the prefrontal cortex (PFC)[55] and dorsal raphe (DR)[56], and ERβ in the hypothalamus and BNST of mice[57]. However, the specific roles of sex steroid hormone receptors in the EWcp likely fluctuate based on changes in circulating steroid hormone levels across the estrous cycle and require greater elucidation. Of note, sex differences in levels of both Ucn1 and CART have been reported in the EW of human suicide victims[58], and studies in rodents show baseline sex differences in these genes plus nesfatin-1[59–61] expression in the EW, suggesting the expression of these peptides may be sexually dimorphic in nature. Interestingly, we have shown here that disrupting circulating ovarian hormones did not alter the ability of EWcp *Ghsr* knockdown to reduce alcohol consumption in female mice.

The EWcp also expresses a number of feeding and arousal related peptide receptors that have gained interest in alcohol research as novel targets for treatment, including receptors for orexin/hypocretin[62,63], leptin[64,65], and ghrelin[23,39]; however, their specific roles within the EWcp have not been characterised. Previous studies have shown increased expression of *Ghsr* mRNA in the EWcp of alcohol preferring male mice (C57BL6J), compared to a non-preferring strain (DBA/2 J)[11]. Our data highlight a sex specific action of *Ghsr* in the EWcp in driving excessive alcohol consumption, whereby ghrelin-induced drinking and effects of viral-mediated knockdown of *Ghsr* in the EWcp were specific to female mice and driven by interactions with peptidergic cells. Moreover, in female mice, *Ghsr* expression positively correlated with alcohol consumption, highlighting the possibility of a direct relationship between *Ghsr* expression and alcohol intake in females. This is in line with previous reports that show sex differences in circulating levels of ghrelin in abstinent[46–48], and alcohol dependent individuals[66], and sensitivity to ghrelin induced consummatory behaviour[44].

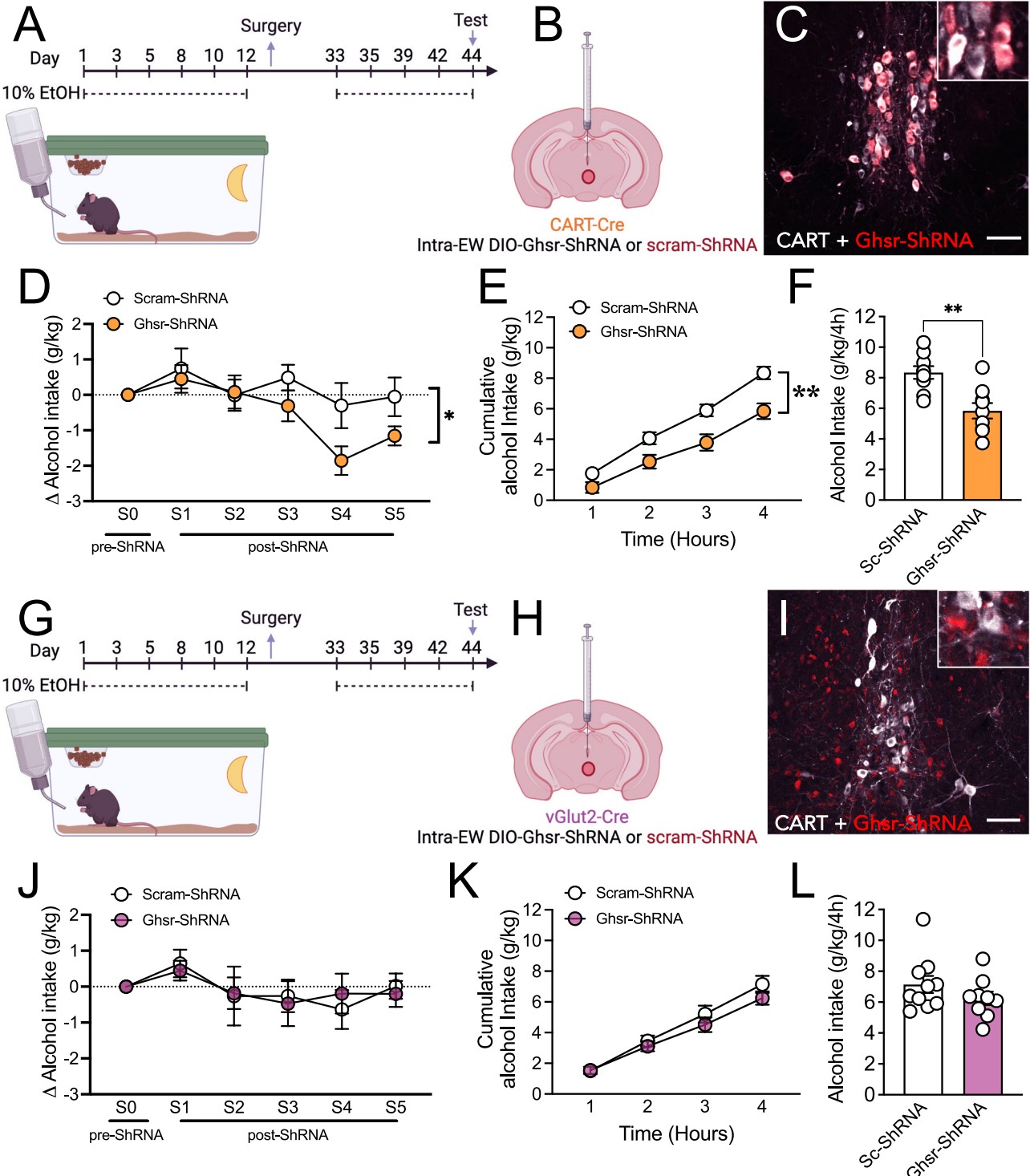

**Fig. 5 | Ghsr acts via EW peptidergic cells to regulate alcohol consumption in female mice. A** Schematic of experimental timeline, **B** details of viral approach and **C** representative image of viral transduction of EW peptidergic cells in the iCART-Cre mouse. **D** GHSR knockdown on EW$^{CART}$ cells reduced alcohol consumption following surgery (main effect of treatment, $p = 0.0082$, but no significant *post hoc* differences; n's=9 Sc-ShRNA, 9 Ghsr-ShRNA), **E** cumulative alcohol intake (1 h $p = 0.2717$, 2 h $p = 0.0812$, 3 h $p = 0.0247$, 4 h $p = 0.0063$; n's=9 Sc-ShRNA, 9 Ghsr-ShRNA) and **F** total alcohol intake in the 4-h session ($p = 0.0015$; n's=9 Sc-ShRNA, 9 Ghsr-ShRNA). **G** Schematic of experimental timeline, **H** details of viral approach and **I** representative image of viral transduction of EW cells in the vGlut2-Cre mouse.

**J** GHSR knockdown on EW$^{vGlut2}$ cells did not alter alcohol consumption following surgery (main effect treatment $p > 0.9588$; n's=10 Sc-ShRNA, 9 Ghsr-ShRNA), **K** cumulative alcohol intake ($p = 0.8302$; n's=10 Sc-ShRNA, 9 Ghsr-ShRNA) or **L** total alcohol intake in the 4-h session ($p = 0.7767$; n's=10 Sc-ShRNA, 9 Ghsr-ShRNA). Data expressed as mean ± SEM. Panel **D, E, J, K** analysed by two-way ANOVA, with Bonferroni *post hoc* for significant main effects, panel **F, L** analysed by two-tailed unpaired students *t*-test. *$p < 0.05$, **$p < 0.01$. Scale bar=200 μm. Full statistics in Table S3 and source data are provided as a Source Data File. Created in BioRender. Walker, L. (2025) https://BioRender.com/a16v552 & https://BioRender.com/o80z071 [Agreement #RA27UETRDV & RA27UETZKU].

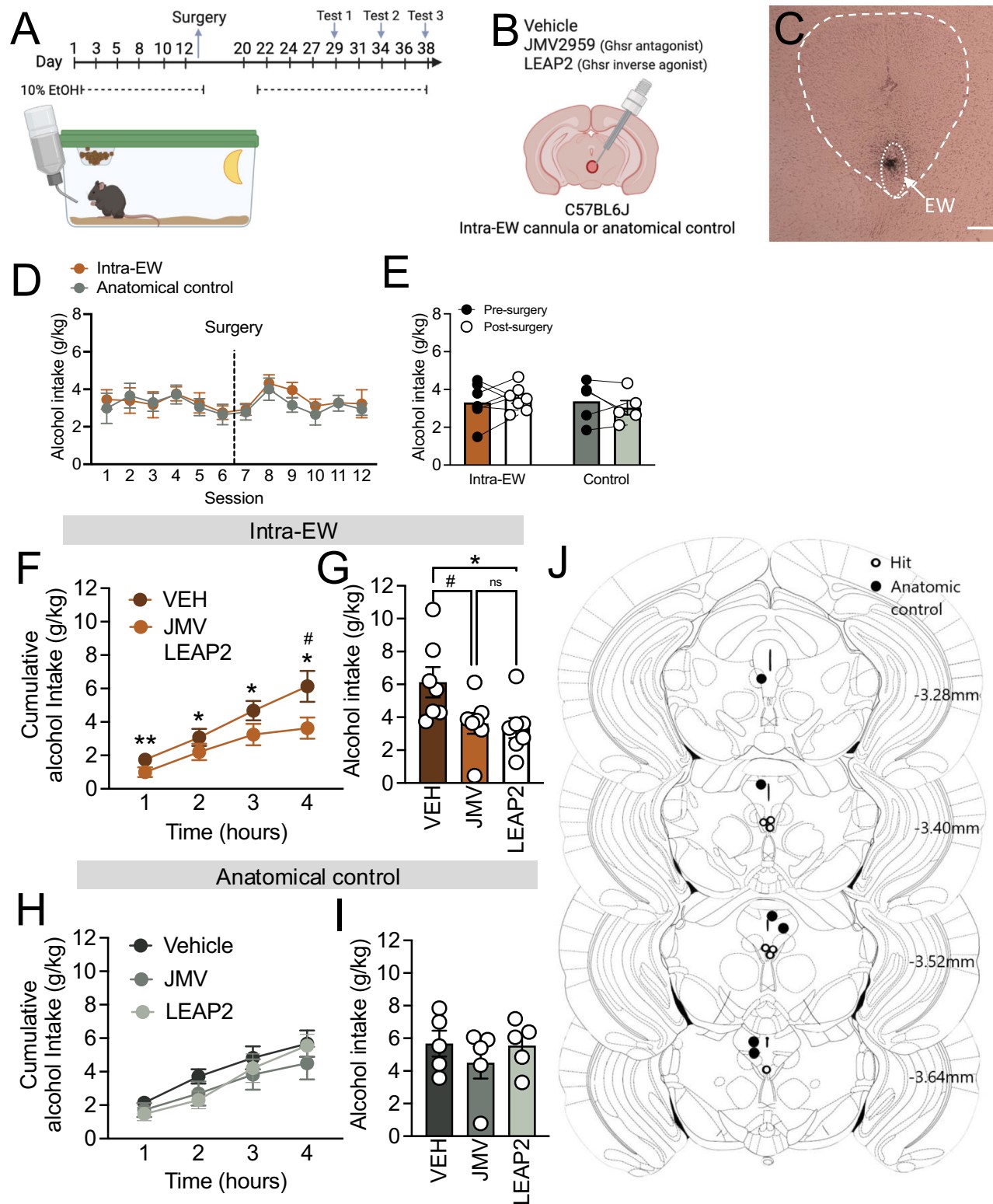

Little is known about the expression, distribution and function of GHSR in the EWcp of both male and female mice. We identified *Ghsr* mRNA expression on both peptidergic and glutamatergic EWcp populations. Our findings differ from previous reports in *Ghsr*-GFP reporter mice suggesting expression overlapped >90% with the peptidergic marker Ucn1[23]; however, this may be due to incomplete reporter expression, or expression levels required to drive recombination and reporter expression in a transgenic mouse line. Indeed, our RNAscope data highlight more dense expression of *Ghsr* mRNA on peptidergic cells, however a large proportion of glutamatergic cells also express *Ghsr* mRNA at lower, but detectable levels. Further, female mice showed a trend towards a greater number of *Ghsr* expressing cells, and *Ghsr* mRNA in the EWcp. Finally, while bath application of ghrelin increased firing rate of peptidergic EWcp cells in both female and male mice, the magnitude of change observed in female mice was significantly higher, suggesting peptidergic cells in the EWcp of female mice are more sensitive to ghrelin. This is in line with the ability of GHSR to couple to Gαs, and increase intracellular calcium

**Fig. 6 | GHSR acts through ligand dependent and independent mechanism to drive alcohol consumption in female mice. A** Schematic of experimental timeline, **B** details of cannulation approach, and **C** representative cannula placement in the EW. **D** No difference in alcohol consumption was observed across training (main effect of treatment $p = 0.6379$; n's = 5 anatomic control, 7 intra-EW) or **E** post-surgery in mice implanted with a cannula in the EW or anatomical controls (intra-EW $p = 0.5725$, anatomical controls $p = 0.4224$; n's=5 anatomic control, 7 intra-EW). **F** LEAP2 and JMV2959 administration within the EW reduced cumulative alcohol consumption during the 4-h test, with LEAP showing significant reduction across all timepoints (1 h $p = 0.0049$, 2 h $p = 0.0279$, 3 h $p = 0.0112$, 4 h $p = 0.0103$; $n = 7$), and JMV2959 reaching significance after 4 h (1 h $p = 0.1430$, 2 h $p = 0.2508$, 3 h

$p = 0.0632$, 4 h $p = 0.0265$; $n = 7$). **G** LEAP2 and JMV2959 administration within the EW reduced total alcohol consumption (LEAP2, $p = 0.0137$; JMV2959, $p = 0.0346$; $n = 7$), however **H** administration of LEAP2 and JMV2959 in anatomical controls (adjacent to the EW) did not alter cumulative (main effect of treatment $p = 0.3797$; $n = 5$) or **I** total alcohol consumption ($p = 0.5029$; $n = 5$). **J** Schematic of cannula placements. Data expressed as mean ± SEM. Scale bar=300 μm. Panel **D, F, H** analysed by two-way ANOVA, Panel **G, J, I** analysed by one-way ANOVA with Bonferroni *post hoc* for significant main effects, Panel **E** analysed by two-tailed paired students t-test. *$p < 0.05$, **$p < 0.01$. Full statistics in Table S3 and source data are provided as a Source Data File. Created in BioRender. Walker, L. (2025) https://BioRender.com/t03f170 [Agreement #VF27UEUH42].

concentrations[24], and with bath application of ghrelin increasing the firing rate of cells across the brain including the Arc[67,68], VTA[49] and central amygdala[69]. However, to our knowledge, direct comparisons between sexes have not been reported until now.

GHSR is also expressed within the neighbouring VTA, on dopaminergic cells, acting to increase dopamine release within the downstream nucleus accumbens (Acb)[42,49,70,71], which can be prevented by peripheral GHSR antagonist administration[42]. While few studies have directly assessed the role of VTA GHSR in alcohol consumption, direct administration of ghrelin into the VTA increased alcohol consumption in male rats[32]. In contrast, with viral mediated knockdown of *Ghsr* in the VTA, we did not see any reductions in alcohol consumption in either sex. Given that our neurochemical characterisation showed expression of *Ghsr* mRNA across both DAT+ and DAT- cells, we further assessed whether specific *Ghsr* knockdown on DAT cells specifically would alter alcohol intake, but failed to see any changes in either sex. While previous reports pharmacologically inhibiting GHSR or selectively knocking down *Ghsr* mRNA in the VTA have not assessed alcohol consumption, pre-treatment with a GHSR antagonist, JMV2959, did not prevent alcohol induced Fos expression in the VTA, only the EW of mice[38]. Together this suggests while ghrelin signalling in the VTA may influence alcohol intake, *Ghsr* knockdown within the VTA does not impact binge drinking in mice.

Of note, GHSR has high constitutive activity and there is much controversy as to how peripheral ghrelin can access the brain[72,73]. We observed that both a GHSR1a antagonist (blocking endogenous ghrelin binding) and inverse agonist (inhibiting the GHSR1a directly) when delivered directly into the EWcp reduced excessive alcohol consumption in female mice. These results suggest endogenous ghrelin, not just constitutive activity of the GHSR1a, contributes to alcohol consumption in female mice. However, previous work using a ghrelin vaccine[39] and blocking peripherally circulating ghrelin[35,74] both failed to see reduction in alcohol intake. Recent evidence suggests physiological state may alter the ability of ghrelin to cross the blood-brain barrier, with chronic stress increasing ghrelin binding in the VTA[75]. Further work is required to elucidate the ability of ghrelin to access the EWcp directly, and the profile of JMV2959, with a report suggesting it may have some inverse agonist properties[76]. Of note, we observed a faster response with LEAP2, but the JMV2959 had a similar effect when measured at the end of the 4-h drinking session. The dynamic differences in the time course of effect on alcohol consumption may be explained by differences in direct actions on receptor signalling by the inverse agonist, compared to blockade of ghrelin-induced receptor activity[26,77] or the specific pharmacokinetics of JMV2959. Recent studies have shown systemic administration of GHSR1a antagonists reduces alcohol consumption in both sexes of mice; however, systemic administration of LEAP2 (inverse agonist) failed to have any effect in either sex[39]. Likely attributed to the undetermined capacity of LEAP2 to penetrate the brain[25], given the same study showed central administration of LEAP2 reduced alcohol consumption[39], which has recently been replicated in rats[78]. Interestingly, LEAP2 and JMV2959 only reduced consumption in high, but not low drinking mice[39]. Given female mice drink alcohol at greater levels than male mice[6], the central

actions of GHSR may be more specific to the excessive consumption of alcohol, contributing to the greater effect observed in female mice within this study.

The EWcp peptidergic neurons send dense projections throughout subcortical regions in the brain including the periaqueductal grey (PAG), lateral septum, dorsal raphe (DR), striatum, BNST, hypothalamus and central nucleus of the amygdala (CeA)[17,18,20]. Many of these regions that have been implicated in mediating sex differences in alcohol consumption (e.g. DR[79], striatum[80,81], BNST[82], CeA[6]). EW[CART] neurons projecting to the DR have been recently implicated in modulating serotonergic activity and anxiety-like behaviours in mice[83]. Further, EW inputs to the Acb core are activated by alcohol consumption[80]; however, this did not differ between sexes. While the EW presents a node in which sex differences in binge drinking may be mediated, further studies to understand the downstream circuitry driving sex differences in alcohol consumption emanating from the EW are warranted.

In summary, we highlight a neurobiological mechanism that underlies the relationship between ghrelin and alcohol consumption. We identify the EWcp as a locus where ghrelin/GHSR1a signalling at peptidergic cells mediates excessive alcohol consumption specifically in female mice. Collectively, our data build upon a growing literature suggesting sex differences in ghrelin/GHSR1a actions in the brain and elucidate mechanisms underpinning sex differences in excessive alcohol consumption.

## Methods
### Animals
C57BL6J mice ($n = 150$) were obtained from ARC (Animal Resources Australia, WA, Australia) at ~6 weeks of age. Breeding stock of inducible CART-Cre knock in mice (iCART-Cre, $n = 45$) kindly donated from Professor Herbert Herzog (Garvan Institute, Sydney,[84,85]), Slc17a6[tm2(cre)Lowl]/J (vGlut2-Cre, $n = 16$) obtained from Zane Andrews (Monash University) originally from the Jackson Laboratory (Bar Harbour, ME, USA, stock # 016963), and Slc6a3[tm1(cre)Xz]/J (DAT-Cre, $n = 26$), kindly donated from Laura Jacobson (Florey Institute) originally from the Jackson Laboratory (stock # 020080) were bred in house at the Florey Institute of Neuroscience and Mental Health by crossing either heterozygous male and female mice or homozygous male mice with C57BL6J female mice. Heterozygous CART-Cre offspring were used for experimentation. Homozygous iCART-Cre mice were crossed with B6.Cg-Gt(ROSA)26Sor[tm14(CAG−TdTomato)Hze]/J (*Ai14*) (Bar Harbour, ME, USA, stock # 007914) to produce iCART-*Ai14* offspring ($n = 14$). Mice were acclimatised to a reverse light cycle (lights off 7.00–19.00 AEST) and single housed in open top cages to allow accurate measurement of fluid and food consumption. All mice had *ad libitum* access to food (laboratory chow, Barastoc) and water except where detailed. All studies were performed in accordance with the Prevention of Cruelty to Animals Act (2004), under the guidelines of the National Health and Medical Research Council (NHMRC) Council Code of Practice for the Care and Use of Animals for Experimental Purposes in Australia (2013) and approved by The Florey Animal Ethics Committee (#21-063).

## Genotyping

iCART-Cre, vGlut2-Cre, DAT-Cre and iCART-Ai14 mice had mutants identified using polymerase chain reaction (PCR) procedures provided by the supplier (refs. 84,85; Jackson Laboratories) (Transnetyx, Cordova, TX, USA).

## Stereotaxic surgery

For surgeries, mice were anesthetized with 4% isoflurane and placed in a stereotaxic frame (Stoelting) and maintained under 1-2% isoflurane for the duration of the surgery. The skull was exposed, and burr holes were made with a drill over the coordinates of the EW (M/L +1.00, A/P −3.40, D/V −4.70, 15 degree angle), or VTA (M/L ± 0.55, A/P −3.0, D/V −4.6). For viral surgeries a Nanoject III (Drummond Scientific, Broomall, PA, USA) was used to deliver 100nL into the EW or 150 nL into the VTA (50 nL/min) of an AAV virus. For DREADD experiments heterozygous iCART-cre mice received AAV2-hsyn-DIO-hM4Di-mcherry or control vector AAV2-hsyn-DIO-mcherry (Addgene #44362-AAV2, # 50459-AAV2). For EW Ghsr knockdown C57BL6J mice received AAV2-hsyn-Ghsr-ShRNA-mCherry or scramble control, AAV2-hsyn-scramble-ShRNA-mCherry, while targeted deletion was achieved using Cre dependent viruses (AAV2-hSyn-DIO-Ghsr-ShRNA-mCherry or AAV2-hSyn-DIO-scramble-ShRNA-mCherry) administered into heterozygous iCART-Cre, vGlut2-Cre or DAT-Cre mice (Vector Biolabs, Malvern, PA, USA).

To allow direct drug administration to the EW, mice underwent stereotaxic cannula implantation. Mice were anaesthetized and prepared as per viral injections. Next, two holes were drilled into the skull and screws (Mr Specs. Parkdale, Australia) were inserted to anchor the cannula to the skull. A small hole was drilled into the skull, through which 26 G stainless-steel guide cannulas cut 3.5 mm below the pedestal (PlasticsOne, Roanoke, VA, USA) were implanted into the EW (M/L +1.00, A/P −3.40, D/V −3.20 mm on a 15 degree angle)[86]. Cannula were fixed to the skull using dental cement (Vertex-Dental) and a dummy, cut projecting 1 mm beyond the cannula tip was inserted (PlasticsOne). Mice received meloxicam (3 mg/kg, i.p.) for analgesia, the antibiotic, baytril (3 mg/kg, i.p.) and saline (10 ml/kg, s.c.) at the end of surgery and were monitored for 5 days following surgery.

## Ovariectomy

Ovariectomy was performed in adult C57BL6J female mice as previously described[6]. Briefly, mice were anesthetised with isoflurane (5% induction; 2% maintenance). The surgical site was cleaned and shaved before a small incision was made along the midline of the lower back, with the skin and underlying muscle separated. An incision was made superiorly through the muscle wall after identifying the ovarian fat pad. The ovary was isolated and excised through this incision. A single suture closed the incision in the muscle wall, and sutures and wound clips used to close the lower back incision. The SHAM surgery was identical to that of the ovariectomy surgery, except the ovary was not disturbed or removed. Following suturing, betadine antiseptic was applied over the sutures, and mice were administered with analgesic (meloxicam; 3 mg/kg⁻¹, s.c.), antibiotic (baytril; 3 mg/kg⁻¹, s.c.), and saline (10 ml/kg⁻¹, s.c.) to facilitate recovery. Mice were given at least 7 days to recover, with body weights and health monitored before recommencing binge drinking training.

## Drugs

LEAP2 (37-76) (5 µg, Phoenix Pharmaceuticals #075-58) or JMV2959 (10 µg, MedChemExpress, #HY-U00433A)[87] diluted in 10% DMSO in 0.9% sterile saline. Ghrelin (BOC Sciences, 1 mg/kg i.p.) was diluted in 0.9% sterile saline. CNO (10 mg/kg) was diluted in 0.05% DMSO in 0.9% sterile saline. All drugs were administered at 10 ml/kg. To activate the CART promoter controlled by the Cre-recombinase gene, and therefore induce specific expression of viruses into EW$^{CART}$ cells, tamoxifen (80 mg/kg, 10 ml/kg, i.p.) (Sigma-Aldrich Pty Ltd, Sydney, NSW, Australia) dissolved in 10% absolute ethanol and 90% sunflower oil was administered to CART-Cre mice (mCherry/DREADD male and female mice, or Sc-ShRNA/Ghsr-ShRNA female mice) for 5 consecutive days from surgery, mice were given at least 10 days before resuming alcohol consumption.

## Drug microinfusion

Intra-EW infusions of Vehicle, LEAP2 or JMV2959 were made using 40 cm polyethylene connectors (PlasticsOne) attached to 1 µL syringes (SGE Analytical Science, Ringwood). Mice were gently held while dummies were removed, and injectors inserted into the cannula. Compounds were infused (0.25 µL/min, 0.5 uL total) by an automated syringe pump (Harvard Apparatus, Holliston, USA). The injectors were left in place for ~2 min after infusion. Mice were habituated to the equipment prior to testing and each mouse received both drugs and vehicle treatment in a randomised and counterbalanced manner. Following behavioural testing to assess cannula positioning, methylene blue (0.5 µL) was infused and animals were anaesthetized with pentobarbitone (100 mg/kg, i.p. Virbac, Milperra, Australia) before decapitation. Brains were collected, frozen over super-cooled isopentane, sectioned (40 µm) on a cryostat (Leica, Leica Microsystems) and mounted onto SuperFrost Plus slides (Menzel-Glaser). After drying, slides were counterstained with Neutral Red solution (Sigma-Aldrich) for 2 min and cleared through a series of ethanol (50%, 70%, 90%, 100%) and X-3B (Olichem Pty Ltd.) washes before cover-slipping with SafteyMount mounting media (Sigma-Aldrich). An investigator blinded to treatment groups and behavioural outcomes performed injection site validations.

## Binge drinking

We adapted the SHAC (scheduled high access consumption) and DID (drinking in the dark) procedures as previously described[6]. Mice had voluntary access to 10% (v/v) ethanol or 5% (w/v) sucrose diluted in tap water, 3 h after the dark cycle began, 3 days per week (Monday, Wednesday, Friday) for 2 h[6]. For DREADD experiments, mice received surgery prior to training and underwent 10 sessions of training before testing. For OVX surgeries, mice were given at least 1 week recovery before access to alcohol was returned. For shRNA and cannulation studies, mice were trained for 6 sessions prior to surgery and given at least 4 sessions post-surgery before testing. To assess ghrelin induced drinking, mice were given access to alcohol in their light phase (3 h after the light cycle began, 3 days per week (Monday, Wednesday, Friday) for 2 h).

Given that ghrelin levels peak in the dark phase, to assess ghrelin-induced drinking, access to alcohol was given during the light phase, when ghrelin levels are low[88]. Mice had access to alcohol 3 h into the light cycle, for 2 h, 3 times weekly during training.

Mice were tested in an extended 4-h drinking session with consumption measured hourly. Mice were given another session to ensure consumption returned to baseline, before testing with the counterbalanced treatment. For DREADD studies CNO was administered 1 h prior to testing. For ghrelin-induced drinking, ghrelin was administered 20 min prior to test onset, and food was removed for the duration of the test. Mice were weighed before binge drinking sessions and ethanol and sucrose consumption in grams per kilogram (g/kg) was calculated using the volume consumed in millilitres (ml) and body weight of each mouse in grams (g). Potential spillage during training and test sessions were assessed using an empty cage housed on the same rack as mice and subtracted from intake (ml) before calculating g/kg intake. For alcohol intake, calories were calculated by g intake x 7 (calories per gram pure alcohol). For food intake, calories were calculated by g intake x 3.01 (calories per gram) (Barastoc).

### Other behaviours in hM4Di mice

During DREADD inhibition test sessions, food consumption was assessed by weighing standard chow at each timepoint after CNO administration. In a separate test, saccharin preference was assessed in a 4 h two bottle choice test where mice had access to either saccharin (0.01%) or water. Preference and intake were measured[6]. To assess anxiety-like behaviours a 10 min light-dark box was employed as previously described[6]. Finally given the implication of the EW in body temperature regulation[21,89], we assessed core temperature pre- and post-CNO administration at an ambient temperature of $23 \pm 1\,°C$ in gently restrained mice using a thermocouple probe (ID-Tech-Bioseb, France; 0.71 mm diameter)[90].

### qPCR

Mice were euthanised via cervical dislocation following behavioural testing. Brains were immediately removed and quickly frozen over liquid nitrogen until sectioning. The EW and VTA of each animal were dissected using needles with inner diameters 1.0-1.5 mm. RNA extraction and analysis were performed as previously described[91,92]. Briefly, RNA from brain tissue was extracted using the QIAGEN Qiazol + RNeasy Plus Micro Kit according to the manufacturer's protocol. Total RNA (300 ng) was then reverse-transcribed into cDNA using SuperScriptII qRT-PCR kit (Invitrogen, USA). SYBR Powerup PCR kit (Qiagen) was used in 10 µL reactions containing 4 µL cDNA, and 1 µL 20 nmol/µL primer mixture. qPCR was performed with ViiA7(Applied Biosystems, USA), followed by melt-curve analysis. The qRT-PCR programme was set to 2 min at 50 °C, 2 min at 95 °C, and 15 s at 95 °C and 1 min at 60 °C (40 cycles). Reactions were performed in 3 technical replicates along with a no template and water control. Primers (Table S1) for were designed using Primer3 v. 0.4.0 software (http://frodo.wi.mit.edu/cgi-bin/primer3/primer3_www. cgi; Whitehead Institute for Biomedical Research, USA) based on coding DNA sequences acquired from GenBank (National Center for Biotechnology Information (NCBI)). Primer specificity was verified (BLAST Interface, NCBI). Relative mRNA expression was analysed with respect to the geometric mean of housekeeper genes *Actin* and *Hprt* (hypoxanthine phosphoribosyltransferase 1) ($\Delta$Ct). $\Delta\Delta$Ct method was used to compare expression between Ghsr-shRNA and Scram-ShRNA controls of each sex. For OVX/SHAM Ghsr-ShRNA, female Scram-ShRNA controls were used to compared expression to OVX Ghsr-ShRNA and SHAM Ghsr-ShRNA samples.

### Immunohistochemistry

Mice were anaesthetised (pentobarbitone, 80 mg/kg, i.p.) and transcardially perfused with 15 mL PBS (0.1 M, pH 7.4) followed by 30 mL 4% PFA in PBS, decapitated, brains removed and post-fixed (1 h) in 10 mL of perfusion solution. For Fos expression after binge drinking, mice were perfused directly after their 2 h alcohol binge session. For viral injection sites, perfusions were not timed with specific behaviour. Post fixation, all brains were incubated in 30% sucrose in PBS (10 mL) at 4°C overnight, before being frozen over liquid nitrogen and stored at −80°C until sectioning. Coronal sections (40 um) were cut on a cryostat (Leica Biosystems) at −18°C and stored in PBS-azide in a 1 in 4 series. Fluorescent immunohistochemistry was performed to examine the co-localisation of either CART and Fos, CART and viral reporter expression, or viral expression alone[93,94]. Briefly, sections were blocked in 10% normal donkey serum (NDS) and 0.5% tritonX-100 (TX-100) in PBS, for 1 h at RT. Sections were then incubated in a primary antibody solution containing rabbit anti-CART (1:2000; Phoenix Pharmaceuticals, #H-003-62) and Goat anti-Fos (1:1000, Santa-Cruz, SC-52-G), or Chicken anti-RFP (Rockland #600-901-379) with 2% NDS in 0.1 M PBS containing 0.1% TX-100 overnight at RT. Sections were washed 3 ×5 min in PBS and incubated for 2 h at room temperature in donkey anti-rabbit AlexaFluor488 (1:400; Life Technologies, #A-21206) and donkey anti-Chicken AlexaFluor594 (Jackson Immunology #703-585-155), or donkey anti-Goat AlexaFluor594 (1:400; Life

Technologies, #A-11012). Sections were finally washed (3 ×5 min) in PBS, mounted on microscope slides and coverslipped with fluorescence mounting medium (DAKO, Australia).

For electrophysiology slices tissue was washed in PBS for 3 ×5 min before antigen retrieval was undertaken. Tissue was incubated in wells of buffer (10 mM Sodium Citrate, 0.05% Tween20, pH 6.0) in an oven at 90°C for 3 h, then left to cool to room temp for 30 min. Tissue was washed in PBS for 3 ×5 min before blocked for 1 h at room temp in 10% NDS + 0.3% TX-100 in PBS. Goat anti-CART h/m/r antibody (R&D systems, #AF163) at 1:1000 in 3% NDS + 0.3% TX-100 in PBS was next incubated with the tissue for 48 h at 4°C with gentle agitation. Tissue was washed in PBS for 3 ×5 min before secondary antibody incubation. Streptavidin HRP 488 (#S11223, 1:500) and donkey anti-Goat Alexa-Fluor594 (#A11058, Lot 1975275, 1:400), in 10% NDS + 0.3% TX-100 + PBS were incubated with tissue for 2 h at RT. Finally, tissue was washed in PBS for 3 ×5 min before mounting on slides with DAKO mounting media and imaging.

### In situ hybridisation

To examine *Ghsr* expression across CART and vGlut2 cells in the EW and DAT cells in the VTA, mice underwent cervical dislocation, brains were rapidly extracted and frozen for 20 secs on dry ice-cooled 2-methylbutane (Bacto Laboratories, NSW, Australia). Brains were stored at -80 °C until use. Coronal sections of the EW and VTA (16 µm) were cut in a 1/6 series using a cryostat (Leica Biosystems, NSW, Australia) and mounted directly onto Super Frost Plus slides (Fisher Scientific, NH, USA). Slides were stored at -80 °C until use. The RNAscope Multiplex Fluorescent Reagent Kit (Advanced Cell Diagnostics) was used to detect *Ghsr*, *Slc17a6* (vGlut2) and *Cartpt* in the EW of C57BL6J mice or *Ghsr* and *Cre* (DAT) in the VTA of DAT-Cre mice[95]. Slides were fixed in 4% w/v paraformaldehyde for 15 mins at 4 °C, before rinsing in 0.1 M PBS (pH 7.4) followed by dehydration in increasing concentrations of ethanol (50%, 70%, 100% and 100% v/v) for 5 min each at RT. Slides were then air-dried for 10 min at RT and a hydrophobic barrier was drawn around the brain sections. Sections were then protease treated (pre-treatment 4) at RT for 5 min. Next, sections were rinsed in distilled water followed by probe application. The target probes used were, *Ghsr* (#426148), *Cartpt* (#432008-C2), *Slc17a6* (#319171-C3), *Cre* (#312281-C2). Slides were first incubated with probes at 40 °C for 1 h. Following this, slides were incubated with preamplifier and amplifier probes (AMP1, 40 °C for 30 min; AMP2, 40 °C for 15 min; AMP3, 40 °C for 30 min), followed by incubation in fluorescently labelled probes to select a specific combination of reporters associated with each channel (AMP4 Alt A to detect *Ghsr* in Alexa-488, *Slc17a6* in Atto 550 channel and *Cartpt* in Atto 647 for EW and AMP4 Alt A to detect *Cre* in Alexa-488, *Ghsr* in Atto 550 channel in the VTA). Finally, sections were incubated for 20 s with DAPI and coverslipped with DAKO Fluorescent Mounting Medium (North Sydney, NSW, Australia).

### Microscopy

A LSM780 Zeiss Axio Imager 2 confocal laser scanning microscope (Carl Zeiss AG, Jena, Germany) with a 40 X objective was used to take all images of the EW (pixel size 0.10 um). FISH analysis was quantified from at least 3 sections per mouse (-Bregma -3.5 mm[86]; using image J (National Institutes of Health; RRID:SCR_003070). For a cell to be deemed positive for the expression of a given gene, five or more mRNA puncta needed to be present[91,94]. A LSM900 Zeiss Axio Imager 2 confocal laser scanning microscope was used to take images of sections used for electrophysiology and imaged at 10 x magnification.

### Electrophysiology

Mice were deeply anesthetized using isoflurane, decapitated and brains were rapidly removed and mounted in a slice chamber

containing chilled cutting solution (125 mM choline chloride, 20 mM D-Glucose, 0.4 mM CaCl$_2$·2H$_2$O, 6 mM MgCl$_2$·6H$_2$O, 2.5 mM KCl, 1.25 mM NaH$_2$PO$_4$ and 26 mM NaHCO$_3$). Slices were cut using a vibratome (Leica VT1200 S), and incubated in a holding chamber with oxygenated artificial cerebrospinal fluid (125 mM NaCl, 10 mM D-Glucose, 2 mM CaCl$_2$·2H$_2$O, 2 mM MgCl$_2$·6H$_2$O, 2.5 mM KCl, 1.25 mM NaH$_2$PO$_4$, and 26 mM NaHCO3) initially at 32 °C for 30 min and then room temperature for at least 30 min before being transferred to the recording chamber. Whole-cell patch-clamp recordings were obtained at 30 °C from visually identified fluorescent CART neurons in the EW. Borosilicate glass electrodes (4–7 MΩ) were filled with an internal solution comprising 125 mM K-gluconate, 5 mM KCl, 2 mM MgCl2·6H2O, 10 mM HEPES, 4 mM ATP-Mg, 0.3 mM GTP-Na, 10 mM phosphocreatine, 0.1 mM EGTA, and 0.2% biocytin, with a pH of 7.2 and osmolarity of 290 mOsm. Baseline data were captured during aCSF bath application, and this was followed by continuous perfusion of aCSF containing Ghrelin (1 μM), with cell recordings commencing after 5 mins. Current- and voltage-clamp recordings were performed using an Axon Multiclamp 700B amplifier (Molecular Devices), Digidata 1440 digitiser (Molecular Devices), and pCLAMP version 10 software. Data were sampled at 50 kHz with a low-pass filter at 10 kHz.

All slice electrophysiology data were analysed using Clampfit version 10.7. Passive membrane properties were calculated from holding potentials of −70 mV and −50 mV using −10 mV test pulses. All other slice electrophysiology data were collected in current clamp. Bridge balance and pipette capacitance neutralisation were manually applied throughout current-clamp experiments. Gap-free recordings (30 s recording time) in the absence of holding current were conducted to measure resting potential and spontaneous firing properties such as firing frequency (Hz). Morphological characteristics were analysed from the first firing event that occurred during the 30 Is recording time. Threshold was defined as the voltage at which the rising membrane potential slope exceeded 20 mV/ms. Amplitude was defined as the AP peak relative to a normalised baseline averaged from 0 ms to 10 ms before threshold. Rise time was defined as the time from 10% to 90% of peak. Width was measured at 50% of peak height. Decay time was defined as the time from 100% to 50% of peak.

### Statistical analysis

The effects of Treatment or Genotype across Time were analysed using an analysis of variance (ANOVA), with repeated measures where appropriate. *Post-hoc* Bonferroni corrections were performed when statistical significance was achieved. For analysis of the effect of Treatment, Genotype or Sex only on behaviour, gene expression or electrophysiological properties, data were analysed using two-tailed Student's *t*-test. Pearson correlations were used to compare behaviour to gene expression data. Details of specific statistical tests undertaken are outlined in Table S3.

### Reporting summary

Further information on research design is available in the Nature Portfolio Reporting Summary linked to this article.

## Data availability

Source data are provided with this paper as a Source Data file. Any additional data are available from the corresponding author. Source data are provided with this paper.

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

## Acknowledgements

We thank the Florey Core Animal Services and Florey Microscopy Facility for their assistance. This project was supported by a Jack Brockhoff Foundation grant (LCW), National Health and Medical Research Council (NHMRC) Ideas grant (2002830, LCW), NHMRC Emerging Leader Fellowship (2008344, LCW), NHMRC synergy grant (2009851, AJL) and Australian Research Training Programme Scholarships (XJM, LTU). We acknowledge support from the Victorian State Government Operational Infrastructure Scheme.

## Author contributions

A.P. Data curation; Formal analysis; Investigation; Writing – original draft; Writing – review & editing. X.J.M. Data curation; Formal analysis; Investigation; Writing - review & editing. P.G. Data curation; Formal analysis; Investigation; Methodology; Writing - original draft; Writing - review & editing. L.T.U. Investigation; review & editing. R.G.A. Investigation; Writing - review & editing. A.S. Data curation; Formal analysis; Investigation. R.M.B. Conceptualization; Writing - review & editing. F.M.R. Conceptualization; Writing - review & editing. W.J.G. Conceptualization; Writing - review & editing. A.J.L. Conceptualization; Resources; Supervision; Writing - review & editing. L.C.W. Conceptualization; Data curation; Formal analysis; Funding acquisition; Investigation; Methodology; Project administration; Supervision; Writing – original draft; Writing - review & editing.

## Competing interests

The authors declare no competing interests.
