## [Transparent Peer Review file · Nature Communications]

Midbrain ghrelin receptor signalling regulates binge drinking in a sex specific manner

Corresponding Author: Dr Leigh Walker

Version 0:

Reviewer comments:

Reviewer #1

(Remarks to the Author)

This is a very interesting and elegant paper. The authors identified sex differences in EWcp CART-expressing cells driving binge-like alcohol consumption. They tested the hypothesis that GHSR signalling in the EWcp was responsible for these sex specific actions and found sex differences in Ghnr mRNA expression and response to ghrelin in the EWcp. The authors further determined that GHSR signalling on CART-expressing cells in the EWcp drives alcohol consumption through both ligand dependent and independent actions. A few minor comments are listed below.

Introduction

1. The ghrelin/GHSR/alcohol literature is well summarized, but I have a couple of recommendations. It says “exogenous ghrelin levels” but in actuality some studies cited by the authors only measured endogenous ghrelin levels, while other studies administered exogenous ghrelin, typically IP in rodents and IV in humans. It is important to briefly acknowledge this distinction. On an additional note, the authors cite an IV ghrelin study conducted in people with AUD (Biol Psychiatry 2014), they may consider mentioning an additional and more recent IV ghrelin study which was also done in people with AUD (Farokhnia et al., Mol Psychiatry 2018)

2. LEAP2 first appears in the Methods. I recommend adding LEAP2 in the introduction when the authors discuss the ghrelin/GHSR system.

Discussion

3. The authors state that, “These results suggest endogenous ghrelin, not just constitutive activity of the GHSR1a, contributes to alcohol consumption in female mice.” On the other hand, recent work does not seem to support a role of peripheral ENDOGENOUS ghrelin per se in alcohol drinking, for example when a ghrelin vaccine was used (Richardson et al. Neuropharmacology 2023; PMID: 37369277; same study the authors cite as reference 30) and when Jerlhag et al. ACER 2014 (PMID: 24428428) used NOX-B11-2, which binds and neutralizes peripherally circulating endogenous ghrelin. On an additional note, Wenthur et al. Science Reports 2019 (PMID: 30755699) reported that a ghrelin vaccine blunted weight gain in mice, however the same vaccine – unlike JMV2959 - did not reduce cocaine CPP. Both Wenthur et al. and Jerlhag et al. used male animals only, hence further highlighting the importance of the work presented here by the authors. That said, Richardson et al. used both male and female animals. Please consider expanding the Discussion on this complex topic.

Reviewer #2

(Remarks to the Author)

This is a potentially interesting and valuable study that examines sex differences in alcohol consumption that are mediated by neurons in the Edinger-Westphal nucleus. The authors find that CART expressing neurons in the EW nucleus of female mice influence drinking whereas these effects were not observed in males. Further, the effects on drinking seems to be mediated by Ghrelin. In general, the study is well carried out and rigorous. The paper is extremely well-written and clear. The authors bring multiple different approaches to bear on this question

I have two major concerns

1) The current presentation of the statistics is unacceptable. All stats are relegated to a supplemental table. Please include the critical stats along with degrees of freedom and test statistic in the main text. The inclusion of p-values as present is not adequate.

The argument that effects are specific to females are not supported by the statistical approach. To buttress this argument sexes should be included in the same test and sex x treatment interactions should precede all post hoc tests. The tables in the supplement list sexes as assessed separately – therefore we can make no inferences about what works and one sex and not the other.

2) The effects are consistently small in the female mice. The fact that the effects in the female are replicated several times over in female mice, inspires confidence that these are real. However, the consistently small effects sizes makes me question the practical significance of these observations. How meaningful is this small of an alteration in drinking? Does this result in a change in blood alcohol levels?

Minor

In the last 5 years there has been an explosion in the study of sex differences. The language that sex differences are “ignored” may have been true in the past, but it is not now. I think the authors can make a more nuanced point here without diminishing the critical need to study sex differences.

Reviewer #3

(Remarks to the Author)

The manuscript by Pearl et al describes a series of studies showing that ghrelin receptors in the centrally projecting Edinger-Westphal nucleus (EWcp) regulate binge alcohol consumption in female, but not in male mice. The findings in this manuscript are exciting for several reasons. Alcohol use disorder (AUD) is highly prevalent and has devastating consequences on health. However, there are only a few pharmacotherapies available and they have insufficient efficacy. As an additional complication, very few medications have been tested in female subjects. The ghrelin system holds much promise as a potential target for the development of AUD medication, and the focus of the manuscript on this system is very important.

The insufficient availability of medications for AUD may be also due to repeated focus of the field on the same few brain regions that are traditionally thought to regulate alcohol dependence. In contrast, this manuscript’s focus on the EWcp is innovative. The work is thorough and comprehensive. The conclusions are supported by the results. Even just the characterization of electrophysiological responses in EWcp provides highly novel and much needed information.

Moreover, the highlight of this study is the clearly demonstrated and partly unexpected female-specific contribution of EWcp to the regulation of alcohol consumption by ghrelin. These findings would make much needed addition to the alcohol research field, studies on sex differences and overall neurobiology of an understudied neurocircuit. Despite all the positive comments, the manuscript has some deficiencies that mostly concern presentation style, language and experimental details. They are listed below in order of appearance:

Line 32. The second sentence in the abstract has no connection to the first one. The authors say “this circuitry”, but it is not clear what they are talking about. The second half of sentence is awkward and needs to be rewritten.

Line 45. Bingeing- means excessive drinking. To “excessive bingeing” makes little sense.

Line 52. Authors present US statistics on AUD, but this is not clear, and not clear why the focus is on US.

Line 60. “The presence of” in this sentence is not needed.

Line 61. Most of the audience is not familiar with the division of EW into EWcp and EWpg. A sentence stating this is also necessary.

Also here, EWcp stands for centrally projecting EW, not “central projecting EW”.

Line 67. While vGlut2 is as marker of glutamatergic cells, this end of the sentence is awkward. Put “expressing” in front of vGlut2.

Line 68. It is strange that references are not provided to the first part of the sentence.

Line 70. “information flowing” sounds too general of a term.

Line 75. What is the source supporting this information? What is this comparison based on?

Line 105 and/or corresponding Supplementary material. Were the iCART-cre mice heterozygous? What was the breeding scheme?

Line 131 and/or corresponding Supplementary material. Were the levels of housekeeping transcripts the same between groups and sexes? How was expression of target genes normalized to the housekeeping genes?

Line 171 and/or Supplementary material. It is not clear when and in which mice the body temperature was assessed.

Line 177. Figure L. Calories derived from alcohol is a useful measure, but it is not clear how it was calculated.

Line 257. Figure 3F-K. It is not clear whether sex differences presented for genes EWcp are specific for EW, or also present in other regions. If it is impossible to obtain these data experimentally, data from the literature on other regions should be discussed.

Line 341. More references should be presented to confirm the first statement. Also here, the second part of this sentence is not “in line” with the first one: sensitivity to alcohol, does not imply that this region regulates alcohol drinking. The beginning

of this paragraph needs to be reworked.

Line 349. Add "effect" after "this".

Line 371. Add "receptors for" before "orexin".

Line 375. Add "effects of" before "viral".

Discussion. I understand the space limitations, but some discussion of projections of EW needs to be included.

Reviewer #4

(Remarks to the Author)

This is an interesting paper arguing for the role of central projecting Edinger-Westphal neurons on alcohol binge-drinking of female, but not male mice. They further argue that it is the CART-expressing EWcp neurons that mediate this effect, and that it is driven by the ghrelin receptor, GHSR. They used various techniques to selectively manipulate EWcp neurons in support of their conclusions.

While I believe that the work is of interest, there are several shortcomings that reduces enthusiasm.

Specific comments:

- 1) From a conceptual perspective, it is very difficult to envision that a specific part of the brain, in this case the EWcp neurons, evolved to solely control binge drinking of alcohol in females, and females only. The study begs the question: what is the physiological role of these neurons? They need to provide some evidence that these neurons are relevant for something else but binge drinking of alcohol and only in females. Why do these neurons even exist in males?
- 2) There are parts of the brain that are sexually dimorphic from mouse to human (e.g., MPOA). Does this apply to the EWcp system?
- 3) Cell-selective alteration of neuronal functions may or may not be relevant to exclude the role of other brain areas in physiology (and in this case pathology). They discussed the role of the midbrain dopamine system as a potential player in these by the discovery of others. They need to do studies to show that interference with those (e.g., midbrain dopamine cells) neurons do not impact this behavior in order to make an exclusive statement on the EWcp. It is not a problem if the midbrain system is also involved, and it would provide a broader understanding how the brain works.
- 4) While statistical significance is absent, trends of many of the phenotypes are seen in males as well. That also applies to the assessment of other behaviors, which the authors concluded are not impacted by EWcp neurons. In this regard, it is important to consider that single housing is a major stress for mice regardless of acclimation.
- 5) Does gonadectomy impact the outcome in either sex?
- 6) Is there any difference in Cart and vGLut2 neurons in terms of activation when comparing males with females? I think it would help understand better the study if the authors showed the activation of the different neurons in males and females are. For example, does alcohol activate a higher percentage of Cart neurons in females than in males? Or the differences in alcohol intake are due just to the fact that ghrelin has a bigger impact on the activity of each single neuron?
- 7) They need to discuss tamoxifen administration in the different cohorts and how that may differ between males and females.

Reviewer #5

(Remarks to the Author)

I co-reviewed this manuscript with one of the reviewers who provided the listed reports as part of the Nature Communications initiative to facilitate training in peer review and appropriate recognition for co-reviewers.

Version 1:

Reviewer comments:

Reviewer #1

(Remarks to the Author)

Reviewer #2

(Remarks to the Author)

No further comments, my concerns have been addressed adequately.

Reviewer #3

(Remarks to the Author)

The manuscript remains an important breakthrough in our understanding of neurocircuitries regulating excessive alcohol consumption. Authors definitively demonstrate involvement of ghrelin receptors of the centrally-projecting Edinger-Westphal nucleus in regulation of this behavior in female, but not male, mice. The study is very comprehensive. Identification of this understudied neurocircuit and its role in alcohol drinking is innovative. The finding of its selective role in females is very

impactful, as currently approved treatments for alcohol use disorder have been developed based on studies in male animals. The authors went far beyond addressing all critiques of the original manuscript by performing multiple additional experiments, demonstrating neuroanatomical specificity of the observed effects and its independence from circulating sex hormones. I have no further concerns with this manuscript and congratulate the authors on an impactful and exciting series of studies.

Reviewer #4

(Remarks to the Author)

The authors have made significant modifications to the manuscript to address the inquiries raised. They conducted new experiments that enhance the strength and scientific value of the study. Specifically, they investigated the role of ghr expression in the VTA, ruling out a contribution of midbrain dopamine cells to alcohol consumption. Additionally, they addressed the role of sex hormones by performing experiments in ovariectomized mice. Furthermore, they observed that CART cells are activated at a higher percentage in female mice.

The study has increased in scientific relevance and is now more significant for both specialized and general audiences

Reviewer #5

(Remarks to the Author)

Response to reviewers – Pearl & Maddern et al.

We thank the reviewers for their positive and constructive feedback; we think the revision is a significant improvement over the original submission. We believe we have addressed all of the reviewers concerns in our revision and added substantial additional data to the revised manuscript. Below we provide point-by-point responses and specific changes highlighted in blue text throughout the manuscript.

Reviewer #1:

This is a very interesting and elegant paper. The authors identified sex differences in EWcp CART-expressing cells driving binge-like alcohol consumption. They tested the hypothesis that GHSR signalling in the EWcp was responsible for these sex specific actions and found sex differences in Ghsr mRNA expression and response to ghrelin in the EWcp. The authors further determined that GHSR signalling on CART-expressing cells in the EWcp drives alcohol consumption through both ligand dependent and independent actions. A few minor comments are listed below.

Introduction

1. The ghrelin/GHSR/alcohol literature is well summarized, but I have a couple of recommendations. It says “exogenous ghrelin levels” but in actuality some studies cited by the authors only measured endogenous ghrelin levels, while other studies administered exogenous ghrelin, typically IP in rodents and IV in humans. It is important to briefly acknowledge this distinction. On an additional note, the authors cite an IV ghrelin study conducted in people with AUD (Biol Psychiatry 2014), they may consider mentioning an additional and more recent IV ghrelin study which was also done in people with AUD (Farokhnia et al., Mol Psychiatry 2018).

Our response: We apologise for this omission. We have now included this study within the introduction and clarified endogenous vs exogenous ghrelin (L81-82). Preclinical and clinical studies have highlighted a bidirectional relationship between endogenous ghrelin levels, or exogenous administration with alcohol consumption/craving [28-35].

2. LEAP2 first appears in the Methods. I recommend adding LEAP2 in the introduction when the authors discuss the ghrelin/GHSR system.

Our response: This has now been included within the introduction (L77-78). “GHSR is activated by its cognate ligand, ghrelin, a 28-amino acid peptide secreted from the stomach, and can be inhibited by the endogenous competitive antagonist/inverse agonist liver-expressed antimicrobial peptide 2 (LEAP-2) [24-26].”

Discussion

3. The authors state that, “These results suggest endogenous ghrelin, not just constitutive activity of the GHSR1a, contributes to alcohol consumption in female mice.” On the other hand, recent work does not seem to support a role of peripheral ENDOGENOUS ghrelin per se in alcohol drinking, for example when a ghrelin vaccine was used (Richardson et al. Neuropharmacology 2023; PMID: 37369277; same study the authors cite as reference 30) and when Jerlhag et al. ACER 2014 (PMID: 24428428) used NOX-B11-2, which binds and

neutralizes peripherally circulating endogenous ghrelin. On an additional note, Wenthur et al. Science Reports 2019 (PMID: 30755699) reported that a ghrelin vaccine blunted weight gain in mice, however the same vaccine – unlike JMV2959 - did not reduce cocaine CPP. Both Wenthur et al. and Jerlhag et al. used male animals only, hence further highlighting the importance of the work presented here by the authors. That said, Richardson et al. used both male and female animals. Please consider expanding the Discussion on this complex topic.

Our response: Thank you for your comment. We agree this is a complex topic and have now added greater discussion to this point, including recent studies published (Richardson et al., 2024) during revision of this manuscript (L491-495). “However, previous work using a ghrelin vaccine [39] and blocking peripherally circulating ghrelin [35, 74] both failed to see reduction in alcohol intake. Recent evidence suggests physiological state may alter the ability of ghrelin to cross the blood-brain barrier, with chronic stress increasing ghrelin binding in the VTA [75]. Further work is required to elucidate the ability of ghrelin to access the EWcp directly, and the profile of JMV2959, with a report suggesting it may have some inverse agonist properties [76].”

Reviewer #2:

This is a potentially interesting and valuable study that examines sex differences in alcohol consumption that are mediated by neurons in the Edinger-Westphal nucleus. The authors find that CART expressing neurons in the EW nucleus of female mice influence drinking whereas these effects were not observed in males. Further, the effects on drinking seems to be mediated by Ghrelin. In general, the study is well carried out and rigorous. The paper is extremely well-written and clear. The authors bring multiple different approaches to bear on this question.

I have two major concerns

1) The current presentation of the statistics is unacceptable. All stats are relegated to a supplemental table. Please include the critical stats along with degrees of freedom and test statistic in the main text. The inclusion of p-values as present is not adequate.

*Our response: To achieve this within the space limitations, in addition to the supplementary table with full statistics (**Table S3**), we have now reported all post-hoc p-values (or main effect if appropriate) within text and figure legends and have provided all source data in a source data file. We are happy to include all statistical reports within the main text if necessary, however this would exceed the current word limitations.*

The argument that effects are specific to females are not supported by the statistical approach. To buttress this argument sexes should be included in the same test and sex x treatment interactions should precede all post hoc tests. The tables in the supplement list sexes as assessed separately – therefore we can make no inferences about what works and one sex and not the other.

Our response: We and many others see baseline differences in alcohol consumption between male and female mice (Maddern et al, 2024, <https://doi.org/10.1038/s41386-023-01712-2>; Bloch et al., 2020, <https://doi.org/10.1016/j.alcohol.2020.07.011>; Hwa et al., 2011, <https://doi.org/10.1111/j.1530-0277.2011.01545.x>; Sneddon et al., 2019, <https://doi.org/>

10.1111/acer.13923; Yoneyama et al., 2008, <https://doi.org/10.1016/j.alcohol.2007.12.006>). Therefore, we assessed the effect of ghrelin within each sex separately. However, to address this comment we have reanalysed normalised data to allow us to compare between sexes. As shown in **Fig. S3** (and below) there are clear differences between males and females in their response to the intervention within the EW, but not VTA.

Supplementary Figure 3, related to Figure 1, 2 and Supp 3: EW inhibition specifically reduces alcohol consumption in female mice. (A) Schematic of viral strategy. **(B)** Female C57BL6J mice with hM4Di in the EW showed a specific reduction in Δ alcohol intake when treated with CNO compared to male mice (RM two way ANOVA, sex $F(1, 15) = 7.874$, $p=0.0133$; time $F(3, 45) = 2.977$, $p=0.0414$; no interaction $F(3, 45) = 0.4816$, $p=0.6967$; Bonferroni post hoc female vs. male S1 $p=0.3966$, S2 $p=0.0429$, S3 $p=0.1357$, S4 $p=0.0399$). **(C)** Schematic of viral strategy. **(D)** Female C57BL6J mice with hM4Di in the EW showed a specific reduction in Δ ghrelin-induced alcohol intake when treated with CNO compared to male mice (RM two-way ANOVA, trend towards main effect, sex $F(1, 14) = 3.590$, $p=0.0790$; no effect of session $F(1.477, 20.68) = 1.766$, $p=0.1997$; or interaction $F(2, 28) = 1.003$, $p=0.3797$). **(E)** Schematic of viral strategy. **(F)** Female mice with GhSr-ShRNA injected in the EW showed a specific reduction in Δ alcohol intake compared to male mice (RM two-way ANOVA, main effect of sex $F(1, 12) = 9.582$, $p=0.0093$; no effect of session $F(2.786, 33.43) = 1.446$, $p=0.2479$; or interaction $F(4, 48) = 1.908$, $p=0.1244$; Bonferroni post hoc female vs. male S1 $p=0.9788$, S2 $p=0.2859$, S3 $p=0.3607$, S4 $p=0.0092$). **(G)** Schematic of viral strategy. **(H)** No difference in Δ alcohol intake was observed between sexes

when *Ghsr-ShRNA* was injected in the VTA of C57BL6J mice (RM two-way ANOVA, no effect sex $F(1, 16) = 0.1393$, $p=0.7139$; interaction $F(6, 96) = 1.396$, $p=0.2242$; main effect of session $F(6, 96) = 4.569$, $p=0.0004$). (I) Schematic of viral strategy. (J) No difference in Δ alcohol intake was observed between sexes when *Ghsr-ShRNA* was injected in the VTA of DAT-Cre mice (RM two-way ANOVA, no main effect sex, $F(1, 9) = 0.04053$, $p=0.8449$; session $F(3.188, 28.69) = 0.8519$, $p=0.4829$, or $F(6, 54) = 1.260$, $p=0.2914$). Data expressed as mean \pm SEM; $n = 9$ female DREADD, 8 male DREADD, 8 female ghrelin + DREADD, 8 male ghrelin + DREADD, 7 female ShRNA, 7 male ShRNA, 11 female C57BL6J ShRNA, 7 male C57BL6J ShRNA, 6 female DAT-Cre ShRNA, 5 male DAT-Cre shRNA. Source data are provided as a source data file. Created using Biorender.com

2) The effects are consistently small in the female mice. The fact that the effects in the female are replicated several times over in female mice, inspires confidence that these are real. However, the consistently small effects sizes makes me question the practical significance of these observations. How meaningful is this small of an alteration in drinking? Does this result in a change in blood alcohol levels?

Our response: Most recent and comparable studies systematically administering a GHSR inverse agonist in C57BL6J mice have shown ~35-40% reductions in alcohol consumption in females (Richardson et al., 2023 and Richardson et al., 2024).

Given in female mice we see:

- ~18% reduction in alcohol intake with inhibition of EW CART cells
- ~30% reduction in ghrelin induced alcohol intake with inhibition of EW CART cells
- ~24% reduction in alcohol intake with EW GHSR knockdown
- ~30% reduction in alcohol intake with GHSR knockdown specifically from CART cells
- ~40% reduction in alcohol intake with EW specific JMV administration
- ~45% reduction in alcohol intake with LEAP2 administration in the EW

Overall, these data consistently show a reduction in alcohol consumption in female mice and suggest that a large portion of GHSR's actions in relation to alcohol consumption may be driven through the EW. While we do not suggest the EW acts alone, we do present a novel locus of action.

In terms of blood alcohol levels, it is well established using this paradigm in C57BL6J mice that consumption correlates strongly with blood levels, in both sexes. Indeed, as reported by Rhodes et al., 2005, <https://doi.org/10.1016/j.physbeh.2004.10.007> "Consumption of ethanol (g/kg) was a significant linear predictor of BEC" and this relationship held over both 2 hours and 4 hours. Therefore, we can be confident that blood alcohol levels would decline by a similar proportion as the g/kg intake.

Minor

In the last 5 years there has been an explosion in the study of sex differences. The language that sex differences are "ignored" may have been true in the past, but it is not now. I think the authors can make a more nuanced point here without diminishing the critical need to study sex differences.

Our response: We agree with the reviewer and have rephrased our language to communicate this point in the abstract (L28-29), "Rates of risky drinking are continuing to rise, particularly in

women, yet sex as a biological variable *has only recently gained traction as a critical factor*” and the introduction(L50-52) “However, sex as a biological variable *has only recently gained traction as a critical factor, with most preclinical research and drug development identifying and testing therapies primarily in male subjects [7].*”

Reviewer #3:

The manuscript by Pearl et al describes a series of studies showing that ghrelin receptors in the centrally projecting Edinger-Westphal nucleus (EWcp) regulate binge alcohol consumption in female, but not in male mice. The findings in this manuscript are exciting for several reasons. Alcohol use disorder (AUD) is highly prevalent and has devastating consequences on health. However, there are only a few pharmacotherapies available and they have insufficient efficacy. As an additional complication, very few medications have been tested in female subjects. The ghrelin system holds much promise as a potential target for the development of AUD medication, and the focus of the manuscript on this system is very important. The insufficient availability of medications for AUD may be also due to repeated focus of the field on the same few brain regions that are traditionally thought to regulate alcohol dependence. In contrast, this manuscript’s focus on the EWcp is innovative. The work is thorough and comprehensive. The conclusions are supported by the results. Even just the characterization of electrophysiological responses in EWcp provides highly novel and much needed information.

Moreover, the highlight of this study is the clearly demonstrated and partly unexpected female-specific contribution of EWcp to the regulation of alcohol consumption by ghrelin. These findings would make much needed addition to the alcohol research field, studies on sex differences and overall neurobiology of an understudied neurocircuit. Despite all the positive comments, the manuscript has some deficiencies that mostly concern presentation style, language and experimental details. They are listed below in order of appearance:

Line 32. The second sentence in the abstract has no connection to the first one. The authors say “this circuitry”, but it is not clear what they are talking about. The second half of sentence is awkward and needs to be rewritten.

Our response: Amended in text (L29-30) “An emerging yet understudied potential component of the circuitry *driving alcohol consumption* is the central projecting Edinger-Westphal...”

Line 45. Bingeing- means excessive drinking. To “excessive bingeing” makes little sense.

Our response: Amended in text “excessive alcohol *consumption*” (L43)

Line 52. Authors present US statistics on AUD, but this is not clear, and not clear why the focus is on US.

Our response: We have clarified this point in the introduction (L47-49). “*Whilst men have historically had higher rates of alcohol use, misuse and alcohol use disorder (AUD) compared to women, statistics from the U.S suggest these rates have converged significantly over recent decades, primarily driven by an increase in risky drinking and AUD rates in women [1-3].*”

Line 60. “The presence of” in this sentence is not needed.

Our response: Amended (L56)

Line 61. Most of the audience is not familiar with the division of EW into EWcp and EWpg. A sentence stating this is also necessary. Also here, EWcp stands for centrally projecting EW, not “central projecting EW”.

Our response: Amended and expanded in text (L61-63) *“The evolutionarily conserved EW is separated into two main divisions, the preganglionic (EWpg), a population of cholinergic neurons that project to the ciliary ganglion, controlling oculomotor coordination, and the centrally projecting (EWcp), which consists of densely clustered neurons that project through the CNS [13-16].”*

Line 67. While vGlut2 is as marker of glutamatergic cells, this end of the sentence is awkward. Put “expressing” in front of vGlut2.

Our response: Amended (L65-66) (*expressing vGlut2*)

Line 68. It is strange that references are not provided to the first part of the sentence.

Our response: Amended (L68)

Line 70. “information flowing” sounds too general of a term.

Our response: Amended in text (L69-71) *“Furthermore, while a role for the glutamatergic EWcp in regulating alcohol consumption was recently identified [21], a model for how the peptidergic EWcp cells regulate alcohol consumption remains incomplete.”*

Line 75. What is the source supporting this information? What is this comparison based on?

Our response: Our comparison based on mRNA analysis from Zigman 2006, <https://doi.org/10.1002/cne.20823>. We apologise for the omission and have now included this reference (L78).

Line 105 and/or corresponding Supplementary material. Were the iCART-cre mice heterozygous? What was the breeding scheme?

Our response: This has been clarified within the methods (L531-536). *“Breeding stock of inducible CART-Cre knock in mice (iCART-Cre, n = 45) kindly donated from Professor Herbert Herzog (Garvan Institute, Sydney, [84, 85], Slc17a6^{tm2(cre)Lowl}/J (vGlut2-Cre, n = 16) obtained from Zane Andrews (Monash University) originally from the Jackson Laboratory (Bar Harbor, ME, USA, stock # 016963), and Slc6a3^{tm1(cre)Xz}/J (DAT-Cre, n = 26), kindly donated from Laura Jacobson (Florey Institute) originally from the Jackson Laboratory (stock # 020080) were bred in house at the Florey Institute of Neuroscience and Mental Health by crossing either heterozygous male and female mice or homozygous male mice with C57BL6J female mice. Heterozygous CART-Cre offspring were used for experimentation.”*

Line 131 and/or corresponding Supplementary material. Were the levels of housekeeping transcripts the same between groups and sexes? How was expression of target genes normalized to the housekeeping genes?

Our response: As shown below, no difference was observed in housekeeper genes between sexes or treatment groups (two-way ANOVA, main effect of sex, $F(1, 22) = 0.4119$, $P=0.5276$; treatment $F(1, 22) = 0.02387$, $P=0.8786$; interaction $F(1, 22) = 0.07307$, $P=0.7894$; $n = 5-7/\text{group}/\text{sex}$). We apologise for not making this clearer in the original manuscript, analysis has been clarified in the methods section (L562-566). “Relative mRNA expression was analysed with respect to the geometric mean of housekeeper genes Actin and Hprt (hypoxanthine phosphoribosyltransferase 1) (ΔCt). $\Delta\Delta\text{Ct}$ method was used to compare expression between Ghsr-ShRNA and Scram-ShRNA controls of each sex. For OVX/SHAM Ghsr-ShRNA, female Scram-ShRNA controls were used to compared expression to OVX Ghsr-ShRNA and SHAM Ghsr-ShRNA samples.”

Line 171 and/or Supplementary material. It is not clear when and in which mice the body temperature was assessed.

Our response: Body temperature was measured in mice with EW DREADD expression pre- and 1 hour post CNO administration. We have clarified this in the methods (L545-546) and added a timeline in Fig. S2 to illustrate. Added to text: “we assessed core temperature pre- and post-CNO administration at an ambient temperature of $23\pm 1^\circ\text{C}$ in gently restrained mice using a thermocouple probe (ID-Tech-Bioseb, France; 0.71 mm diameter) [90].”

Line 177. Figure L. Calories derived from alcohol is a useful measure, but it is not clear how it was calculated.

Our response: Detail of calculation added within the methods (L534-537). “For alcohol intake, calories were calculated by g intake $\times 7$ (calories per gram pure alcohol). For food intake, calories were calculated by g intake $\times 3.01$ (calories per gram) (Barastoc).”

Line 257. Figure 3F-K. It is not clear whether sex differences presented for genes EWcp are specific for EW, or also present in other regions. If it is impossible to obtain these data experimentally, data from the literature on other regions should be discussed.

Our response: Sex and developmental differences in ER binding and ER α / ER β expression in the rodent brain. While we find no data for the EW specifically, we have expanded our discussion on this point (L432-435). “Sex and developmental differences in ER binding have been reported within the hypothalamus, bed nucleus of the stria terminalis (BNST) and ARC of rodents [54]. Further, sex differences in ER α expression have been reported within the prefrontal cortex (PFC) [55] and dorsal raphe (DR) [56], and ER β in the hypothalamus and BNST of mice [57].”

Line 341. More references should be presented to confirm the first statement. Also here, the second part of this sentence is not “in line” with the first one: sensitivity to alcohol, does not imply that this region regulates alcohol drinking. The beginning of this paragraph needs to be reworked.

Our response: We agree with the reviewer. And now with additional experiments to show EW CART cells have greater activation than male counterparts (see **Fig. 1**), we have restructured this section (L408-414). “EWcp peptidergic cells are strongly activated by alcohol [9, 12, 13], in line with this we showed activation of EWcp peptidergic cells in response to binge drinking, and that selective chemogenetic inhibition of this population of cells reduces alcohol consumption. Interestingly, while alcohol activates the EWcp, specific comparisons of activation between sexes had not been assessed previously. However, the EWcp has recently been implicated as a connector hub linking regions activated by excessive alcohol consumption in female, but not male mice [53]. In line with this, we show while alcohol activates peptidergic EW cells in both sexes, a greater activation is observed in females, and this correlates to alcohol consumption..”

Line 349. Add “effect” after “this”.

Line 371. Add “receptors for” before “orexin”.

Line 375. Add “effects of” before “viral”.

Our response: We have amended these within text (L419, L445, L449).

Discussion. I understand the space limitations, but some discussion of projections of EW needs to be included.

Our response: We have expanded this point in the discussion (L509-517) “The EWcp peptidergic neurons send dense projections throughout subcortical regions in the brain including the periaqueductal gray (PAG), lateral septum, dorsal raphe (DR), striatum, BNST, hypothalamus and central nucleus of the amygdala (CeA) [17, 18, 20]. Many of these regions that have been implicated in mediating sex differences in alcohol consumption (e.g. DR [79], striatum [80, 81], BNST [82], CeA [6]). EW^{CART} neurons projecting to the DR have been recently implicated in modulating serotonergic activity and anxiety-like behaviours in mice [83]. Further, EW inputs to the Acb core are activated by alcohol consumption [80]; however, this did not differ between sexes. While the EW presents a node in which sex differences in binge drinking may be mediated, further studies to understand the downstream circuitry driving sex differences in alcohol consumption emanating from the EW are warranted.”

Reviewer #4:

This is an interesting paper arguing for the role of central projecting Edinger-Westphal neurons on alcohol binge-drinking of female, but not male mice. They further argue that it is the CART-expressing EWcp neurons that mediate this effect, and that it is driven by the ghrelin receptor, GHSR. They used various techniques to selectively manipulate EWcp neurons in support of their conclusions. While I believe that the work is of interest, there are several shortcomings that reduces enthusiasm.

Specific comments:

1) From a conceptual perspective, it is very difficult to envision that a specific part of the brain, in this case the EWcp neurons, evolved to solely control binge drinking of alcohol in females, and females only. The study begs the question: what is the physiological role of these neurons? They need to provide some evidence that these neurons are relevant for something else but binge drinking of alcohol and only in females. Why do these neurons even exist in males?

Our response: We have expanded our introduction to highlight the role of the EW regulating both sex independent (energy metabolism and anxiety responses) and sex dependent functions (preparatory nest building in female mice) – (L66-67). “...and these neurons are critical for regulating energy homeostasis [19] and anxiety responses [18] across sexes, and preparatory nesting in pregnant female mice [17].”

2) There are parts of the brain that are sexually dimorphic from mouse to human (e.g., MPOA). Does this apply to the EWcp system?

Our response: This is unclear – however, the conserved nature of the EW nucleus (both EWpg and EWcp) across human, non-human primates, cats, ferrets and rodents (Horn et al., 2008 – <https://doi.org/10.1111/j.1749-6632.2009.03856.x>) would suggest this could be possible. Further, in terms of sex steroid hormone receptors, female macaques express both estrogen and progesterone receptors in peptidergic neurons of the EW (Lima et al., 2008 <https://doi.org/10.1016/j.brainres.2008.05.078>; Sanchez et al., 2010 <https://doi.org/10.1016/j.neuroscience.2010.08.059>).

3) Cell-selective alteration of neuronal functions may or may not be relevant to exclude the role of other brain areas in physiology (and in this case pathology). They discussed the role of the midbrain dopamine system as a potential player in these by the discovery of others. They need to do studies to show that interference with those (e.g., midbrain dopamine cells) neurons do not impact this behavior in order to make an exclusive statement on the EWcp. It is not a problem if the midbrain system is also involved, and it would provide a broader understanding how the brain works.

Our response: We have now conducted corresponding studies that show *Ghsr* knockdown in the VTA does not significantly alter alcohol consumption in mice of either sex. RNAscope analysis revealed *Ghsr* expression on ~67% of dopamine (DAT+) and 33% on other cells in the VTA. Therefore, we also induced *Ghsr* knockdown in the VTA of DAT-Cre mice, which again failed to reduce alcohol consumption in our protocol. This is now added as **Fig. S4** (& below).

Supplementary Figure 4, VTA GHSR knockdown does not alter binge drinking in male or female mice. (A) Representative image of *Ghnr* expression with *DAT (cre)* in the VTA. (B) *Ghnr* mRNA was expressed in 67% *Cre+* (*DAT*) cells in the VTA, but also on 33% *DAT*-negative cells. (C) Schematic of viral strategy. (D) *Ghnr*-ShRNA reduced *Ghnr* mRNA expression in the VTA (unpaired *t*-test, $t=4.450$, $df=32$, $p<0.0001$). (E) No significant difference was observed in female mice post *shRNA* knockdown during training (Two-way ANOVA, no effect of treatment $F(1, 17) = 0.1594$, $p=0.6947$; interaction $F(11, 187) = 0.4879$, $p=0.9093$; main effect of session $F(4.962, 84.35) = 2.883$, $p=0.0191$; Bonferroni post hoc showed no difference to *S0*

in Sc-ShRNA or Ghnr-ShRNA, p 's > 0.1) or **(F)** cumulative intake during test (Two-way ANOVA, no effect of treatment $F(1, 17) = 0.0008908$, $p = 0.9765$; interaction $F(3, 51) = 0.2372$, $p = 0.8700$; main effect of time $F(3, 51) = 140.3$, $p < 0.0001$). **(G)** No significant difference was observed in male mice post shRNA knockdown during training (Two-way ANOVA, no effect of treatment $F(1, 14) = 2.262$, $p = 0.1548$; interaction $F(11, 154) = 1.335$, $p = 0.2102$; main effect of session $F(4.523, 63.32) = 7.010$, $p < 0.0001$; Bonferroni post hoc compared to S0, Sc-ShRNA S0 vs. S3 $p = 0.0039$, S0 vs. S10 $p = 0.0640$; Ghnr-ShRNA S0 vs. S6 $p = 0.0092$, S0 vs. S10 $p = 0.0669$) but a trend in **(H)** cumulative intake during test (Two-way ANOVA, treatment $F(1, 13) = 3.150$, $p = 0.0993$; main effect of time $F(3, 39) = 98.46$, $p < 0.0001$; no interaction $F(3, 39) = 0.2024$, $p = 0.8941$). **(I)** Schematic of viral strategy. **(J)** Representative image of Ghnr-ShRNA in the VTA of DAT-Cre mice. **(K)** No significant difference was observed in female DAT-Cre mice post shRNA knockdown during training (Two-way ANOVA, treatment $F(1, 9) = 0.001528$, $p = 0.9697$; or interaction $F(6, 54) = 1.275$, $p = 0.2845$; main effect of session $F(2.405, 21.65) = 4.730$, $p = 0.0153$; Bonferroni post hoc showed no difference to S0 in Sc-ShRNA or Ghnr-ShRNA, p 's > 0.1) or **(L)** cumulative intake during test (Two-way ANOVA, treatment $F(1, 17) = 0.0008908$, $p = 0.9765$; interaction $F(3, 51) = 0.2372$, $p = 0.8700$; main effect time $F(3, 51) = 140.3$, $p < 0.0001$). **(M)** No significant difference was observed in male DAT-Cre mice post shRNA knockdown during training (Two-way ANOVA, no effect treatment $F(1, 10) = 0.1883$, $p = 0.6735$; session $F(3.158, 31.58) = 1.615$, $p = 0.2038$; or interaction $F(6, 60) = 0.4416$, $p = 0.8481$) or **(N)** cumulative intake during test (Two-way ANOVA, no effect of treatment $F(1, 10) = 0.1027$, $p = 0.7552$; interaction $F(3, 30) = 0.2577$, $p = 0.8552$; main effect time $F(3, 30) = 73.20$, $p < 0.0001$). Data expressed as mean \pm SEM. $n = 7$ female C57BL6J Sc-ShRNA, 12 female C57BL6J Ghnr-ShRNA, 9 male C57BL6J Sc-ShRNA, 7 female C57BL6J Ghnr-ShRNA; $n = 5$ female DAT-Cre Sc-ShRNA, 6 female DAT-Cre Ghnr-ShRNA, 7 male DAT-Cre Sc-ShRNA, 5 male DAT-Cre Ghnr-ShRNA. Source data are provided as a source data file. Created using Biorender.com

4) While statistical significance is absent, trends of many of the phenotypes are seen in males as well. That also applies to the assessment of other behaviors, which the authors concluded are not impacted by EWcp neurons. In this regard, it is important to consider that single housing is a major stress for mice regardless of acclimation.

Our response: We acknowledge that single housing is an added stressor to animals. Our mice are housed in open top cages next to littermates to reduce this stress (clarified in methods L544) "... single housed in open top cages to allow accurate measurement of fluid and food consumption." We also see greater alcohol consumption in female C57BL6J mice when group housed between binge sessions, suggesting increased alcohol consumption in female mice is not driven by single housing stress (See below, Walker lab, unpublished, RM two-way ANOVA Main effect sex $F(1, 46) = 63.28$, $P < 0.0001$, main effect time $F(11, 506) = 10.09$, $P < 0.0001$, interaction $F(11, 506) = 4.085$, $P < 0.0001$. $n = 26$ female, 22 male mice).

5) Does gonadectomy impact the outcome in either sex?

Our response: We have now conducted an additional experiment to determine whether circulating hormones reduce or enhance the ability of GHSR knockdown to reduce alcohol intake in female mice. Our data show that *Ghsr* knockdown in the EW reduces alcohol intake in female mice independent of OVX, with no difference between OVX and SHAM surgery mice (**Fig. 3** and below).

Figure 3. Circulating sex hormones does not impact the effect of *Ghsr* knockdown in female mice. (A) Schematic of experimental timeline, (B) details of viral approach and (C) qPCR validation of *Ghsr* mRNA expression in the EW following ShRNA knockdown in SHAM ($p=0.0040$) and OVX ($p=0.0075$). (D) Training pre- and post-ShRNA knockdown showed a reduction in alcohol consumption in female mice with SHAM or OVX and ShRNA knockdown compared to their baseline (S0) (SHAM, S0 vs. S1 $p=0.003$, S0 vs. S2 $p=0.0302$, S0 vs. S3 $p=0.0066$, S0 vs. S4 $p=0.0057$, S0 vs. S5 $p=0.3250$, S0 vs. S6 $p=0.0339$; OVX, S0 vs S1 $p=0.0005$, S0 vs. S2 $p=0.0050$, S0 vs. S3 $p=0.0017$, S0 vs. S4 $p=0.0008$, S0 vs. S5 $p=0.0018$, S0 vs. S6 $p=0.0009$), but no difference between OVX and SHAM groups ($p=0.3802$). No difference in alcohol consumption was also observed in SHAM or OVX *Ghsr*-ShRNA treated mice during an extended 4-hour test in both (E) cumulative ($p=0.9530$) or (F) total alcohol intake ($p=0.5458$). (G) A reduction in uterine weights were observed in OVX mice compared to SHAM ($p=0.0012$). Data expressed as mean \pm SEM, $n=10$ female/group. Panel C analysed by one-way ANOVA, Panel D & E analysed by Two-way ANOVA. Bonferroni post hoc used when significant main effects observed for ANOVAs, Panel F & G analysed by unpaired student t-test. ** $p < 0.01$; \$ $p < 0.05$, \$\$ $p < 0.01$, \$\$\$ $p < 0.001$ compared with S0 in SHAM *Ghsr*-ShRNA mice; # $p < 0.05$, ## $p < 0.01$, ### $p < 0.001$ compared with S0 in OVX *Ghsr*-ShRNA mice. Full statistics in Table S3 and source data are provided as a source data file. Created using Biorender.com

6) Is there any difference in Cart and vGLut2 neurons in terms of activation when comparing males with females? I think it would help understand better the study if the authors showed the activation of the different neurons in males and females are. For example, does alcohol activate a higher percentage of Cart neurons in females than in males? Or the differences in alcohol intake are due just to the fact that ghrelin has a bigger impact on the activity of each single neuron?

Our response: We have conducted an additional experiment to determine how alcohol impacts activation of CART⁺ and CART⁻ cells in the EW nucleus. Our data show CART cells in the EW of male and female mice are activated by alcohol compared to alcohol naïve controls; however, we see a greater % of CART cells activated in female compared to male mice, which also correlates with alcohol intake. This has been added to **Fig. 1A-G** (& below). However, given our neurochemical characterisation, we do also believe that there is greater Ghrelin expression and action of ghrelin within the EW nucleus of female mice.

Figure 1: EW^{CART} cells mediate alcohol binge drinking in female, but not male mice. (A) Schematic of experimental timeline for assessing EWcp activity after binge drinking. **(B)** Female mice consumed more alcohol than male mice ($p=0.0217$). **(C)** Representative image of CART + Fos labelling in the EWcp. **(D)** Female and male mice showed increased %CART+Fos expression in the EWcp after alcohol consumption compared to naïve controls (female $p<0.001$, male $p=0.0152$), with female alcohol mice showing greater %CART+Fos than male counterparts ($p=0.0132$). **(E)** Alcohol consumption showed a positive correlation with %CART+Fos expression in the EWcp ($R^2 = 0.3877$, $p=0.0132$). **(F)** Female, but not male, mice showed increased %Fos+CART-positive expression in the EWcp (naïve vs. alcohol female $p=0.0187$; male $p=0.4232$), but neither sex showed differences in **(G)** %Fos+CART-negative (female $p=0.2222$; male $p=0.2215$) nor differences between sexes in alcohol groups ($p=0.6353$). **(H)** Schematic of experimental timeline for chemogenetic targeting,

7) They need to discuss tamoxifen administration in the different cohorts and how that may differ between males and females.

Our response: We agree, this is an important point – Tamoxifen acts as an estrogen receptor partial agonist. Tamoxifen was only administered in iCART-Cre mice to induce Cre expression and the treatment finished at least 10 days before alcohol access was given post-surgery. While we do not have pre-surgery intake for mice that underwent surgery for mCherry or hM4Di DREADDs, where both male and female mice were included, we do have unpublished data showing tamoxifen administration 2 weeks prior to returning to alcohol in iCART-Cre mice (after control injections) does not alter average alcohol consumption (see below - Unpublished data, Walker lab, paired t-test, female $p=0.9391$; male $p=0.3830$, $n=5/\text{sex}$). We have added more detail to the methods to clarify the timeline in which mice received tamoxifen and resumed access to alcohol (L593-598). “To activate the CART promoter controlled by the Cre-recombinase gene, and therefore induce specific expression of viruses into EW^{CART} cells, tamoxifen (80 mg/kg, 10 ml/kg, i.p.) (Sigma-Aldrich Pty Ltd, Sydney, NSW, Australia) dissolved in 10% absolute ethanol and 90% sunflower oil was administered to CART-Cre mice (mCherry/DREADD male and female mice, or Sc-ShRNA/Ghsr-ShRNA female mice) for 5 consecutive days from surgery, mice were given at least 10 days before resuming alcohol consumption.”